# Simulating the effect of subsurface drainage on the thermal regime and ground ice in blocky terrain, Norway

Cas Renette[1,2], Kristoffer Aalstad[1], Juditha Aga[1], Robin Benjamin Zweigel[1,3], Bernd Etzelmüller[1], Karianne Staalesen Lilleøren[1], Ketil Isaksen[4], Sebastian Westermann[1]

[1]Department of Geosciences, University of Oslo, Oslo, Norway
[2]Department of Earth Sciences, University of Gothenburg, Gothenburg, Sweden
[3]Centre for Biogeochemistry of the Anthropocene, UiO, Oslo
[4]Norwegian Meteorological Institute, Oslo, Norway

*Correspondence to*: Cas Renette (cas.renette@gvc.gu.se)

**Abstract.** Ground temperatures in coarse, blocky deposits such as mountain blockfields and rock glaciers have long been observed to be lower in comparison with other (sub)surface material. One of the reasons for this negative temperature anomaly is the lower soil moisture content in blocky terrain, which decreases the duration of the zero curtain in autumn. Here we used the CryoGrid community model to simulate the effect of drainage on the ground thermal regime and ground ice in blocky terrain permafrost at two sites in Norway. The model setup is based on a one-dimensional model domain and features a surface energy balance, heat conduction and advection, as well as a bucket water scheme with adjustable lateral drainage. We used three idealized subsurface stratigraphies, *blocks only*, *blocks with sediment* and *sediment only*, which can be either *drained* (i.e. with strong lateral subsurface drainage), or *undrained* (i.e. without drainage), resulting in six scenarios. The main difference between the three stratigraphies is their ability to retain water against drainage: while the *blocks only* stratigraphy can only hold small amounts of water, much more water is retained within the sediment phase of the two other stratigraphies, which critically modifies the freeze-thaw behaviour. The simulation results show markedly lower ground temperatures in the *blocks only, drained* scenario compared to other scenarios, with a negative thermal anomaly of up to 2.2 °C. For this scenario, the model can in particular simulate the time evolution of ground ice, with build-up during and after snow melt and spring and gradual lowering of the ice table in the course of the summer season. The thermal anomaly increases with larger amounts of snowfall, showing that well drained blocky deposits are less sensitive to insulation by snow than other soils. We simulate stable permafrost conditions at the location of a rock glacier in northern Norway with a mean annual ground surface temperature of 2.0–2.5 °C in the *blocks only, drained* simulations. Finally, transient simulations since 1951 at the rock glacier site (starting with permafrost conditions for all stratigraphies) showed a complete loss of perennial ground ice in the upper 5 m of the ground in the *blocks with sediment, drained* run, a 1.6 m lowering of the ground ice table in the *sediment only, drained* run and only 0.1 m lowering in the *blocks only, drained* run. The interplay between the subsurface water/ice balance and ground freezing/thawing driven by heat conduction can at least partly explain the occurrence of permafrost in coarse blocky terrain below the elevational limit of permafrost in non-blocky sediments. It is thus important to consider the subsurface

water/ice balance in blocky terrain in future efforts on permafrost distribution mapping in mountainous areas. Furthermore, an
accurate prediction of the evolution of the ground ice table in a future climate can have implications for slope stability, as well
as water resources in arid environments.
**1 Introduction**
Permafrost is defined as ground that remains at or below 0 °C for two or more consecutive years (Van Everdingen, 1998). It
is a common feature in the Arctic and high mountain environments, where permafrost occurs even in mid- and low latitudes
(Gorbunov, 1978). Different permafrost zones are classified based on the aerial extent of permafrost presence. These zones
are: continuous, discontinuous, sporadic and isolated, where the surface in underlain by permafrost in more than 90%, 50-
90%, 10-50% and less than 10% of the land area, respectively (Smith and Riseborough 2002). Snow is an important factor in
governing ground temperatures and permafrost distribution within an area (e.g. Zhang et al., 2001; Zhang 2005; Goodrich,
1982), especially in mountain areas where permafrost is often associated with a shallow snow cover (e.g. Gisnås et al., 2014;
Luetschg et al., 2004). The influence of soil moisture is complicated as it has an impact on the surface energy balance (e.g
Liljedahl et al., 2011), the thermal characteristics of the soil (e.g. Göckede et al., 2017), and freezing/thawing dynamics (e.g.
Hinkel et al., 2001; Hinkel and Outcalt, 1994), which can lead to both lower and higher ground temperatures. Finally, the
thermal and hydrological properties of the subsurface material can strongly influence permafrost distribution. In discontinuous
mountain permafrost terrain, the lowest-lying permafrost areas are frequently found in coarse, blocky terrain (Harris and
Pedersen, 1998). In particular, rock glaciers are frequently found below the general elevation limit of mountain permafrost
(Lilleøren and Etzelmüller 2011).
In Southern Norway, the lower limit of mountain permafrost is estimated between 1600 m a.s.l. in the west to 1000
m a.s.l. in the east (Etzelmüller et al., 2003), while a similar west-east decrease from 800–1000 m a.s.l. to ca. 300 m a.s.l. in
the east is observed in Northern Norway (Gisnås et al., 2017). A first Norway-wide inventory of rock glaciers based on aerial
imagery was published in 2011 (Lilleøren and Etzelmüller, 2011). The density of rock glaciers is lower than in other mountain
permafrost areas which was attributed to a lack of bedrock competence and debris availability as well as to the relative lack of
steep topography above the permafrost limit. While this first inventory suggested that active rock glaciers occur only above
400 m a.s.l., Lilleøren et al. (2022) recently described rock glaciers near sea level in the area of Hopsfjorden, northern Norway,
which feature a limited ice body and are in transition from active to relict. Furthermore, Nesje et al. (2021) presented new
evidence for active rock glaciers in southern Norway well below the permafrost limits established in modelling studies
(Westermann et al., 2013; Gisnås et al., 2017).
Rock glaciers play an important role in the hydrological cycle, especially in arid regions like the Andes, where in
some areas more water is stored in rock glaciers than in glaciers (Jones et al., 2019; Azócar and Brenning, 2010). The open
debris structure can act as a trap for snow and rock glaciers can store significant quantities of ice or liquid water. Rock glaciers
studied in Argentina are an important water resource as they release water mainly during periods of drought (Croce and Milana

2002). Sustained ground ice melt as a response to climate warming threatens this water source. Additionally, melting of ground ice can lead to slope instability (e.g. Gruber and Haeberli, 2007; Saemundsson et al., 2018; Nelson et al., 2001) and damage to infrastructure (e.g. Arenson et al., 2009).

The occurrence of a negative temperature anomaly in coarse, blocky deposits has long been recognized (e.g. Liestøl, 1966). Harris and Pedersen (1998) found a negative temperature anomaly of 4 to 7 °C in blocky terrain relative to adjacent mineral sediment in mountains in Canada and China. They summarized four hypotheses that explain these anomalies: (a) The Balch effect; (b) chimney effect; (c) continuous air exchange with the atmosphere when no continuous winter snow cover is present; and (d) evaporation of water and sublimation of ice in the summer. The first three of these driving mechanisms relate to air movement in the blocks, while the last hypothesis links characteristics of the water/ice balance to lower ground temperatures in blocky terrain. In the Norwegian mountains, Juliussen and Humlum (2008) showed that blockfields featured a negative temperature anomaly of 1.3 to 2.0 °C. They state that convection in the blockfields is of low importance in creating the anomaly, while the effect was mainly attributed to rocks protruding into and through the snow cover which leads to an increased heat transfer through the snow cover. Gruber and Hoezle (2008) presented a simple model for the conductive effect of blocks protruding through the snow cover and showed that the mean annual ground temperature is reduced as a result of a lower thermal conductivity of a blocky layer. Additionally, Juliussen and Humlum (2008) argued that a low soil moisture content in permeable blocky debris (due to subsurface drainage in permeable blocky debris) accelerates active layer refreezing in autumn since less latent heat is liberated compared to soils with higher soil moisture content. Cold winter temperatures can therefore penetrate to deeper layers already in early fall/winter, which may lead to decreased overall winter temperatures. However, in spring, the opposite effect is observed when percolating meltwater refreezes at the bottom of the blocky surface layer, leading to rapid ground warming to 0 °C even in deeper layers (e.g. Juliussen and Humlum, 2008; Hanson and Hoelzle, 2004; Humlum, 1997).

While many of the mechanisms and processes governing the ground thermal regime of blocky terrain are known, a comprehensive quantitative understanding is still lacking. This is particularly relevant for conceptualization in numerical models which generally do not account for the thermal anomaly of blocky terrain. One-dimensional heat flow models have been used in studies to investigate the effect of climate change on permafrost (e.g. Etzelmüller et al., 2011; Hipp et al., 2012) or to model specific processes in mountain permafrost (e.g. Gruber and Hoezle, 2008). Since permafrost presence is generally not visible at the surface, numerical models are often used to estimate the permafrost distribution (Harris et al., 2009). However, as most models neither include a transient representation of the subsurface water and ground ice balance (e.g. Westermann et al., 2013) nor reproduce the thermal anomaly in blockfields (e.g. Obu et al., 2019), the resulting permafrost maps likely show biased ground temperatures and permafrost extent in mountain areas.

The CryoGrid community model (Westermann et al., 2022) is a simulation toolbox that can calculate ground temperatures and water/ice contents in permafrost environments. It largely builds on the well-established CryoGrid 3 model (Westermann et al., 2016) which has been used in e.g. peat plateaus and palsas (Martin et al., 2021), ice-wedge polygons (Nitzbon et al., 2019) and boreal forests (Stuenzi et al., 2021) and has a broad range of applications, including the representation

of lateral drainage regimes (Martin et al., 2019), representation of steep rock walls (Schmidt et al., 2021) and massive ice
bodies. In the following, the CryoGrid community model is referred to as "CryoGrid" for simplicity.
In this study, we present CryoGrid simulations of the coupled heat and water/ice balance for blocky terrain in Norway
and evaluate the impact of the ground stratigraphy and the drainage regime on ground temperatures. The model is set up with
forcing data for two Norwegian permafrost sites, namely a blockfield site in the high mountains in southern Norway and a
rock glacier site near sea level in northern Norway. The employed model scheme does not account for air movement and rocks
protruding the snow cover as the "classic" causes for the negative thermal anomaly of blocky terrain, but is capable of
simulating the seasonal dynamics of the ground ice table in blocky terrain. The goal of the study is to evaluate to what extent
the thermal anomaly in blocky terrain can be simulated by such a comparatively simple scheme which could in principle be
integrated in larger-scale permafrost modelling and mapping efforts. In particular, we investigate the interplay with the
seasonal snow cover and discuss the impact on the permafrost distribution in mountain environments.
**2 Study sites**
**2.1 Juvvasshøe, southern Norway**
Juvvasshøe (61°40 N, 08°22 E, 1894 m a.s.l.) (Fig. 1) is a site located in Jotunheimen in the southern Norwegian mountains,
well above the tree line. A 129 m deep borehole was drilled in August 1999 in the PACE (Permafrost and Climate in Europe)
project (Harris et al., 2001). Continuous data streams from this PACE borehole are available with the exception of a gap
between 21 December 2011 to 24 April 2014. The site is located in an extensive block field on a mountain plateau with sparse
vegetation cover. The bedrock (crystalline rocks, Farbrot et al., 2011) is located at approximately 5 m depth, the first meter
consists of large stones and boulders and the ground below mainly consists of cobbles (Isaksen et al., 2003). Between 2000
and 2004, Isaksen et al. (2007) measured a mean annual air temperature (MAAT) at 2 m height of -3.3 °C. The mean ground
temperature at 2.5 m below the surface during this period was -2.5 °C. The mean annual precipitation was estimated to be
between 800 and 1000 mm. The site is extremely wind-exposed, resulting in a low snow thickness due to wind drift. Hipp et
al. (2012) described a snow depth of less than 20 cm, while the snow thickness in surrounding, lower-lying and less exposed
sites can be up to 140 cm. Isaksen et al. (2007) measured the difference between the mean annual ground surface temperature
and MAAT, which is the surface offset, at exposed and less exposed sites in this area. At sites with a significant snow cover,
the surface offset was more than 2 °C, while at exposed (including Juvvasshøe) sites this offset is generally below 1 °C. The
permafrost thickness at the PACE borehole was estimated to be approximately 380 m (Isaksen et al., 2001), with the lower
permafrost limited at ca. 1450 m a.s.l. (Farbrot et al., 2011). The thickness of the active layer increased from 215 cm in 1999
(Isaksen et al., 2001) to ca. 250 cm in 2019 (Etzelmüller et al., 2020). A weak zero curtain effect suggests a low water content
in the active layer (Isaksen et al., 2007). A warming of 0.2 °C per decade and 0.7 °C per decade in surface air temperature and
ground surface temperature, respectively, occurred between 2000 and 2019 (Etzelmüller et al., 2020).

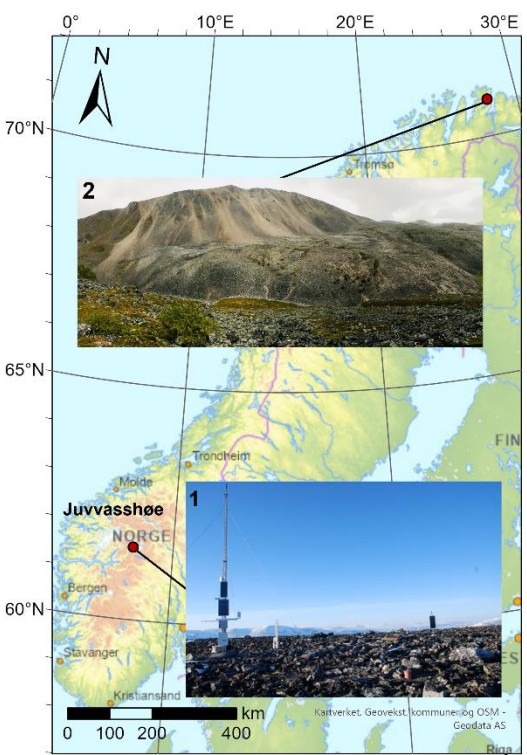


**Figure 1: Location of the two sites in Norway (© Norwegian Mapping Authority). (1) blockfield at Juvvasshøe (1894 m a.s.l.), (2)**
**rock glacier at Ivarsfjorden (60–160 m a.s.l.).**

## 2.2 Ivarsfjorden rock glacier, northern Norway

Ivarsfjorden is a small fjord arm of the larger Hopsfjorden, located on the Nordkinn peninsula in the Troms and Finnmark
county in northern Norway (Fig. 1). Deglaciated around 14–15 cal kyr BP (calibrated kiloyears before present, Romundset et
al., 2011), the peninsula is dominated by flat mountain plateaus of exposed bedrock, *in situ* weathered material and coarse
grained till (Lilleøren et al., 2022), which feature steep slopes towards the sea. The coastal areas of Finnmark have a wet
maritime climate, with mean annual precipitation around 1000 mm (Saloranta, 2012). Lilleøren et al. (2022) describe a MAAT
of 1.6 °C between 2010 and 2019 in the area of the rock glacier, which lies in a southwest-northeast trending valley at an
elevation extent of roughly 60 to 160 m a.s.l.. The mountain at its east (443 m a.s.l.) serves as the source area with rockfall
debris and coarse talus slopes being common. The bedrock in Ivarsfjorden consists of sandstones and phyllites (NGU, 2008).
Sandstones often generate coarse, bouldery material, which is favorable for the formation of rock glaciers (Haeberli et al.,
2006). The rock glacier in Ivarsfjorden is northwest facing and has previously been interpreted as relict (Lilleøren and
Etzelmüller, 2011), but a detailed analysis showed that a limited ice core might still be present (Lilleøren et al., 2022). A
negative MAAT around 100 to 150 years ago is an indication that rock glaciers in this area were likely active at the end of the
Little Ice Age (LIA). Refraction Seismic Tomography surveys indicate a porous air-filled stratigraphy such as blocky talus
deposits at the near-surface at parts of the rock glacier. While observed mean annual ground surface temperatures between
2015 and 2020 are all positive, negative surface temperatures during summer have been observed by a thermal camera at the
front slope of the rock glacier. This is likely an indication of the chimney effect and thus of connected voids that support air
flow.
**3. Methods**
**3.1 The CryoGrid community model**
CryoGrid is a simulation toolbox for ground thermal simulations that can be applied to a wide range of modelling tasks in the
terrestrial cryosphere thanks to its modular structure (see Westermann et al., 2022 for details). It is mainly applied in permafrost
environments, using the finite difference method to transiently simulate ground temperatures. We use a one-dimensional model
column with a domain depth of 100 m (e.g. Westermann et al., 2016; Schmidt et al., 2021) and grid cell sizes increasing with
depth (Fig. 2). The lower boundary condition is provided by a constant geothermal heat flux. The upper boundary results from
solving the surface energy balance, including both radiative and turbulent heat fluxes, as well as the heat flux in the ground.
In order to compute the surface energy balance, atmospheric forcing data are required (Sect. 3.2). To calculate ground
temperatures, both conductive heat transfer following Fourier's law and advection of heat with vertically moving water is
taken into account (Westermann et al., 2022). The freezing characteristic of subsurface water/ice depends on the soil type,
either following Painter and Karra (2014) for sediments, or set to free water (water changes state at 0 °C, Westermann et al.,
2022) for subsurface material with large pores/voids, such as blocky terrain. To define the properties of the subsurface material,
a stratigraphy of volumetric mineral, organic, water and ice contents and the field capacity (the ability to hold water against
gravity) must be provided (Westermann et al., 2022).

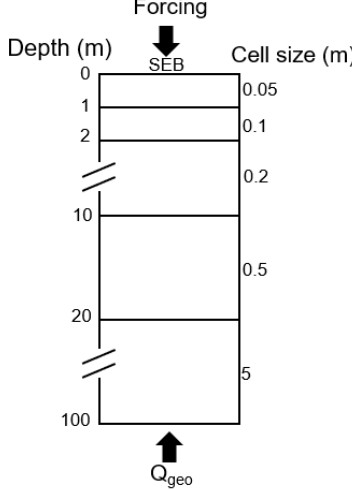

**Figure 2: Schematic of the model grid, indicating cell sizes at different depths and upper and lower boundary conditions. As upper**
**boundary condition, the surface energy balance (SEB) forced by near-surface meteorological data is used. The lower boundary**
**condition is provided by a constant geothermal heat flux, $Q_{geo}$.**

171        For soil hydrology, a gravity driven bucket scheme is used (Westermann et al 2022). Rainfall provided by the model

forcing is added to the uppermost grid cell, while evaporation is determined by the surface energy balance calculations (note
that we consider unvegetated surfaces and thus do not account for transpiration). Water that is in excess of the field capacity
infiltrates downwards until either the water table or a non-permeable layer, such as a frozen grid cell is reached. If all grid cells
are saturated, excess water is removed as surface runoff. We use a one-dimensional model setup, but simulate lateral drainage
of water by introducing a seepage face, i.e. a lateral boundary condition for water fluxes representing flow between the
saturated grid cells of the model domain and a stream channel (or the atmosphere) to which the water can freely flow out from
the subsurface (e.g. Scudeler et al., 2017). Using the elevation of the water table, $z_\mathrm{wt}$ (computed as the elevation of the
uppermost saturated grid cell), lateral water fluxes $F_i^{lat}$ are derived for all saturated unfrozen grid cells $i$ below the water table
(i.e. at elevations $z_\mathrm{i} < z_\mathrm{wt}$ ) as
$$F_i^{lat} = -\mathrm{K}_H \frac{z_\mathrm{wt} - z_\mathrm{i}}{d^{lat}} , \tag{1}$$
where $\mathrm{K}_H$ is the saturated hydraulic conductivity, $d^{lat}$ is the lateral distance to the seepage face and the flux is determined by
the difference between the hydrostatic potential (proportional to $z_\mathrm{wt}$ ) of the water column and the gravitational potential of
free water at the elevation of each cell (proportional to $z_\mathrm{i}$ ). Note that Eq. (1) is an approximation for small changes of the water
table and small outflow fluxes for which the potential in the saturated zone can be approximated by the hydrostatic potential.
The parameter $d^{lat}$ is used to control the strength of the drainage, with small distances resulting in a well-drained column,
while high values lead to suppressed drainage. In this study, we consider the two confining cases with a small and large value
of $d^{lat}$, respectively (Sect. 3.3). In the former, water from rain or ground ice melt is removed rapidly, effectively preventing
the soil water from pooling up, while drainage is negligible in the former, so that the setup corresponds to a classic one-
dimensional model scheme.

191        The snow model used in this study was introduced by Zweigel et al. (2021) and is based on the Crocus snow scheme

(Vionnet et al., 2012) which accounts for snow microphysics and is designed to reproduce a realistic snow pack structure (see
Vionnet et al., 2012 for defining equations; Zweigel et al., 2021 for implementation in CryoGrid). Snowfall is added with
density and microphysical properties derived from model forcing data, in particular air temperature and wind speed. The snow
density evolves due to compaction by the overburden pressure of overlying snow layers, as well as wind compaction and
refreezing of melt- and rainwater (Vionnet et al., 2012). The amount of snowfall from the forcing data can be adjusted by a so-
called *snowfall factor, sf,* with which the snowfall rate from the model forcing is multiplied. With this, the effects of wind-
induced snow redistribution on ground temperatures can be represented at least phenomenologically (Martin et al., 2019),
using *sf* < 1 for areas with net snow ablation and *sf* > 1 for areas with net deposition.

## 3.2 Downscaling of model forcing

The meteorological data used to force the CryoGrid model were generated by applying TopoSCALE, a topography-based downscaling routine (Fiddes and Gruber, 2014), to ERA5 reanalysis data (Hersbach et al., 2020). TopoSCALE is employed in cryosphere applications in complex terrain, including estimating mountain permafrost distribution (Fiddes et al., 2015), snow data assimilation (Aalstad et al., 2018; Fiddes et al., 2019), and downscaling regional climate model output (Fiddes et al., 2022). ERA5 output is provided as interpolated point values on a regular latitude-longitude grid at a resolution of 0.25° at an hourly frequency, both at the surface level, corresponding to Earth's surface as represented in the reanalysis, and at 37 pressure levels in the atmosphere from 1000 to 1 hPa.. We obtained data for the entire reanalysis period, from 1951 to 2019, and converted this into a moving three-hourly average, which is the temporal resolution that the model is run at. As input to TopoSCALE, we obtained from the surface level: 2 m air and dewpoint temperature, 10 m meridional (northward) and zonal (eastward) wind velocity components, surface pressure, constant surface geopotential, incoming longwave radiation, incoming shortwave radiation, and total precipitation. From the pressure levels we acquired: air temperature, specific humidity, zonal and meridional wind velocity components, and dynamic geopotential. For Juvvasshøe we used all levels in the range 900 hPa to 700 hPa, while for the lower elevation Ivarsfjorden rock glacier we used all levels between 900 hPa and 1000 hPa. To account for terrain shading in the downscaling routine, a digital elevation model (DEM) is required, for which we use the mosaic version of the ArcticDEM with a resolution of 32 m (Porter et al., 2018) at both sites. TopoSCALE delivered all meteorological forcing data required to run CryoGrid: near surface air temperature, specific humidity, wind speed, incoming longwave radiation, incoming shortwave radiation, as well as snowfall and rainfall.

## 3.3 Model setup

Three idealized ground stratigraphies are set up in order to investigate the effect of water drainage on the ground thermal regime and ground ice dynamics in blocky terrain. These are referred to as the *blocks only, blocks with sediment* and *sediment only* stratigraphies (Table 1) in the following. The *blocks only* stratigraphy consists of a coarse block layer with 50% porosity and air-filled voids which is assigned low field capacity of 1% (Table 1), i.e. the surfaces of the coarse blocks retain only little water. This idealized stratigraphy is designed to represent an active rock glacier where finer sediments resulting from weathering and erosion processes are transported towards the tongue of the rock glacier. Furthermore, Dahl (1966) observed that blockfields on slopes more often do not contain a fine sediment fraction between the blocks in northern Norway, so that the *blocks only* stratigraphy can also represent active blockfields. The second stratigraphy, *blocks with sediment*, is designed to represent blocky terrain where the voids are filled by finer sediments. This is often observed in blockfields on more flat surfaces, which are more likely to retain finer sediment within their pores (as in Isaksen et al., 2003 and Dahl 1966). We again consider coarse blocks with 50% porosity (as for the *blocks only* stratigraphy), but as the voids are filled with fine sediments (which again are assumed to have 50% porosity), the overall porosity is only 25%. Furthermore, a significantly higher field capacity than for the *blocks only* stratigraphy is assigned as more water can be held in the finer pores of the sediment fraction.

Finally, the *sediment only* stratigraphy serves as a control scenario for a soil without blocks. It contains sediment with 50%
porosity and a high field capacity due to the water holding capacity of the fine-grained sediment material. For all stratigraphies,
bedrock (3% porosity and saturated conditions, e.g. Hipp et al., 2012; Fabrot et al., 2011) is assumed below 5 m depth, which
is in line with observations from Isaksen et al. (2003) at Juvvasshøe. Finally, none of the stratigraphies contain soil organic
matter. We emphasize that the stratigraphies are in qualitative agreement with field observations of air and sediment-filled
block layers in Norway, but the assumed porosities of 50% for both the block layer and the sediments represent idealized
scenarios. However, we perform a sensitivity analysis for different porosity values (Table S1 in the Supplement) to investigate
the impact of this parameter on the simulation results.

**Table 1: Mineral content, porosity, field capacity (all in vol. fraction) and soil freezing characteristic for the three idealized**
**subsurface stratigraphies.**

| Name | mineral | porosity | field capacity | soil freezing characteristic |
|---|---|---|---|---|
| *Blocks only* | 0.5 | 0.5 | 0.01 | Free water |
| *Blocks with sediment* | 0.75 | 0.25 | 0.15 | Free water |
| *Sediment only* | 0.5 | 0.5 | 0.25 | Sand |


For the geothermal heat flux lower boundary condition, a value of 0.05 Wm$^{-2}$ is used, which is a typical value for Norway used
in previous modelling studies (Westermann et al., 2013).
To investigate the effect of subsurface drainage on ground temperatures and ground ice conditions, we distinguish *undrained*
and *drained* scenarios by using two different values of $d^{lat}$ (Eq. 1) for in the idealized stratigraphies. A $d^{lat}$ value of $10^4$ m is
used for *undrained* cases, which emulates conditions at a flat surface, resulting in a to a good approximation one-dimensional
water balance, where only surface water is removed. For the *drained* cases, a $d^{lat}$ value of 1 m is used, which results in well-
drained conditions which are typical in sloping terrain. For the saturated hydraulic conductivity $K_H$, a fixed value of $10^{-5}$ m s$^{-1}$
is used for all stratigraphies, although the true hydraulic conductivities almost certainly differ between stratigraphies.
However, the key parameter controlling lateral water fluxes in Eq. 1 is in reality the "drainage timescale" $K_H/d^{lat}$ [s$^{-1}$], which
is varied by four orders of magnitude between $K_H/d^{lat} = 10^{-5}$ s$^{-1}$ ($d^{lat} = 1$ m, well-drained conditions) and $K_H/d^{lat} = 10^{-9}$ s$^{-1}$
($d^{lat} = 10^4$ m undrained conditions). As the study setup is designed to analyze these two "confining cases", it is sufficient to
only vary $d^{lat}$ and leave $K_H$ constant for simplicity. Further sensitivity tests for $d^{lat}$ and $K_H$ are provided in Table S2 in the
Supplement. With the exception of the *snowfall factor* (see Sect. 3.3.1 to 3.3.3), the parameters in the snow model are kept
constant in all model runs, using a surface emissivity of 0.99, a roughness length of $10^{-3}$ m, a saturated hydraulic conductivity
of $10^{-4}$ m s$^{-1}$ and a field capacity of 0.05 (Westermann et al., 2022). For the ground surface, we used an albedo of 0.15,
emissivity of 0.99, and a roughness length of $10^{-3}$ m.

260        We perform three types of model simulations which differ in their overall purpose. For *validation* runs (Sect. 3.3.1),

we adjust subsurface stratigraphy and *snowfall factor* in order to compare model results with the available field measurements
from the two study sites. *Equilibrium* runs (Sect. 3.3.2) and *transient* runs (Sect. 3.3.3) are designed to explore the sensitivity
of the simulated ground thermal regime towards the three idealized stratigraphies (Table 1) and the two drainage cases. An
overview of the basic settings of the different simulation types is provided in Table 2.

**Table 2: Overview of basic model settings for the different simulation types. A spin-up of subsurface temperatures is achieved by**
**repeated simulations for the spin-up period (until a stable temperature profile is reached), before the actual model run for the**
**simulation period is conducted. "Idealized" stratigraphy and drainage refers to three subsurface stratigraphies (Table 1) combined**
**with two types of drainage conditions. See Sect. 3.3.1 to Sect. 3.3.3 for details.**

| Simulation type | Site | Spin up period | Simulation period | Stratigraphy and drainage | Snowfall factor |
|---|---|---|---|---|---|
| *Validation* | Juvvasshøe | 1951-2010 | 2010-2019 | Best-fit | 0.25 |
|  | Ivarsfjorden | 1951-2016 | 2016-2019 | Best-fit | 1 |
| *Equilibrium* | Juvvasshøe | 2000-2010 | 2000-2010 | Idealized | 0.0, 0.25, 0.5, 0.75, 1.0, 1.5 |
|  | Ivarsfjorden | 1962-1971 | 1962-1971 | Idealized | 0.0, 0.25, 0.5, 0.75, 1.0, 1.5 |
| *Transient* | Juvvasshøe | 1962-1971 | 1951-2019 | Idealized | 0.25 |
|  | Ivarsfjorden | 1962-1971 | 1951-2019 | Idealized | 1 |


**3.3.1 Validation runs**
As a prerequisite for conducting model experiments on ground stratigraphy and drainage (Sects. 3.3.2, 3.3.3), validation runs
are set up to show that the model can reproduce key characteristics of the thermal regime at the two sites in a satisfactory
manner (based on available observations). Furthermore, we use the observations to determine the best-fitting *snowfall factor*
for the two sites which is subsequently used in the transient runs (Sect. 3.3.3). At Juvvasshøe, temperature measurements in a
borehole are available from 2000 to 2019, allowing a comparison at different depths. At the Ivarsfjorden rock glacier site,
observations of ground temperature at deeper depths are lacking, but measurements of near-surface ground temperatures are
available from July 2016 to July 2019 (Lilleøren et al., 2022). These are compared to simulation results to ensure that the
model reproduces the observed surface offset between air and ground surface, largely caused by the winter snow cover (e.g.
Martin et al., 2019; Schmidt et al., 2021). At both sites, the model is ran for the entire period of available forcing data, leaving
at least 60 years for the model spin-up which is sufficient to analyze ground temperatures in uppermost meters of the ground
column.
Manual adjustment of the ground stratigraphy (porosity and thus mineral content) and snowfall factor are performed
until a good fit with daily measurements is achieved. At Juvvasshøe, based on observations of blocks and smaller cobbles with
finer sediments down to the onset of bedrock at a depth of 5 m (Isaksen et al., 2003), the *blocks with sediment* stratigraphy is
used as a starting point to vary porosities until a good fit is achieved. As this site is extremely exposed to wind and most snow
is blown away (Isaksen et al., 2003; Westermann et al., 2013), the snowfall factor is stepwise decreased to values below one
to improve the model performance. At Ivarsfjorden, we considered 11 temperature loggers within the rock glacier outline (Fig.
1d in Lilleøren et al., 2022), of which all except for one are placed on the relict surface of the rock glacier (Fig. 2a in Lilleøren
et al., 2022). On the relict surface, deposition of finer sediment in between blocks is more likely than on the active surface,
due to the lack of movement. Here, the *blocks with sediment* stratigraphy is considered appropriate and used as starting point
for the calibration. At both sites, the root-mean-square-error (RMSE) and bias are calculated in order to provide an objective
measure of the model fit. At Juvvasshøe this was accomplished for daily values at 0.4 m and 2 m depth, while at Ivarsfjorden
the mean daily ground surface temperature of the loggers within the rock glacier outline is used.

### 3.3.2 Equilibrium runs

The goal of equilibrium runs is to investigate the sensitivity of the ground thermal regime towards ground properties and
drainage conditions, using both the *undrained* and *drained* setup for the three idealized stratigraphies (Table 1) which results
in six scenarios. As the heavily wind-affected snow cover is a key source of spatial variability in ground temperatures in the
Norwegian mountains (Gisnås et al., 2014; Gisnås et al., 2016), the model is run for a range of snowfall factors between 0.0
and 1.5 (Table 2) for each scenario. This analysis allows us to identify the magnitude of the thermal anomaly that the subsurface
drainage induces at various amounts of snow, as well as estimate the threshold snow amount for permafrost existence in the
six scenarios. This analysis is performed for equilibrium conditions for 10 year periods of roughly stable climate, which is
iterated three times until a steady state temperature profile of the uppermost 5 meters is established. For Juvvasshøe, the period
2000 to 2010 is selected as the model can be initialized with real-time borehole data. For Ivarsfjorden, the comparatively cold
period 1962 to 1971 is selected as this relatively stable period is the coldest period in the available forcing data and thus the
closest to Little Ice Age climate conditions, when the Ivarsfjorden rock glacier was very likely active (Lilleøren et al., 2022).

### 3.3.3 Transient runs

The goal of the transient runs is to analyze the effect of ground stratigraphies and drainage conditions on the transient response
of ground temperatures and ice tables to climate warming. For this purpose, we perform model simulations from 1951 to 2019,
when air temperatures have increased by more than 1 °C in Norway. To initialize simulations, we perform a model spin-up by
iterating three times over the coldest 10 year period in the forcing data (1962 – 1971) which is sufficient to achieve a stable
ice table. This is the same period as in the equilibrium runs (see Sect. 4), for which it was selected to capture permafrost
conditions at the Ivarsfjorden rock glacier site (see Sect. 4). Thus, the transient runs allow us to analyze the evolution of the
permafrost towards the warming of the recent decades. We only use the best fitting snowfall factor (Table 2), as derived from
the validation runs (Sect. 3.3.1), but again perform simulations for the three idealized stratigraphies and *undrained* and *drained*
conditions. This way, we can evaluate whether different ground stratigraphies or drainage conditions lead to different warming
rates of ground temperatures, as well as different thresholds for permafrost thaw.

## 4. Results

### 4.1 Comparison to in-situ measurements

The results of the *validation runs* at Juvvasshøe are compared with measured daily ground temperatures at the PACE borehole
(Etzelmüller et al, 2020). Figure 3 shows the comparison of measured ground temperatures with modelled temperatures at 0.4
and 2.0 m depth for the best fitting model configuration. The snowfall factor for this model setup is 0.25, i.e. incoming snowfall
is reduced by 75% in order to capture the effect of snow ablation due to wind drift. This resulted in mean annual maximum
snow depths of 34 cm, in broad agreement with observations from the site (Iskasen et al., 2003) and earlier modeling studies
at the site (Westermann et al., 2013). The subsurface stratigraphy for this model configuration is highly similar to the *blocks*
*with sediment* stratigraphy, but with a slightly lower porosity of 0.2 (i.e. a volumetric mineral content of 0.8). This would for
example correspond to blocks and cobbles with a porosity of 0.4 (0.5 for *blocks with sediment*), filled with fine sediments with
a porosity of 0.5 (and field capacity 0.25), which is plausible given the broad characteristics of the observed borehole
stratigraphy (Isaksen et al., 2003). This configuration used *drained* conditions, although differences with *undrained* conditions
are minimal for this stratigraphy. For daily temperatures at 0.4 m depth, the RMSE and bias are 2.1 °C and -0.6 °C, respectively,
while they are 1.2 °C and -0.7 °C at 2 m depth. There is a mismatch in the timing of spring temperatures at 2 m depth in several
years, for which modelled temperatures increase later than measured values. This is likely a result of differences in the snow
melt, as the snowpack dynamics resulting from wind redistribution is not completely captured by the snowfall scaling with a
constant snowfall factor (e.g. Martin et al., 2019). Furthermore, the uppermost 1 m contain large stones and boulders, while
the layer below is characterized by smaller stones and cobbles (Isaksen et al., 2003), so that a ground stratigraphy with two
layers in the uppermost 5 m may further improve the performance of the simulations.

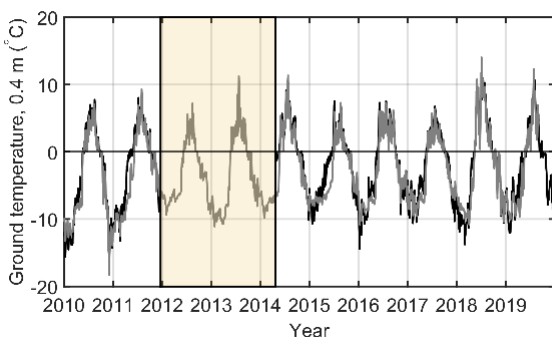

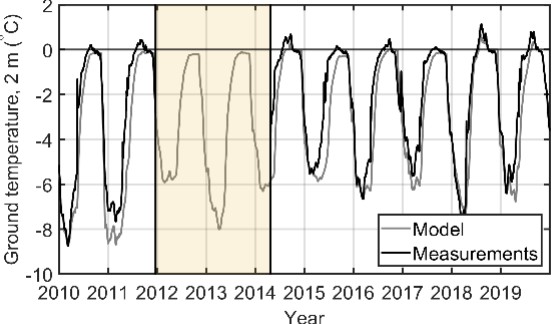

**Figure 3: Modelled and measured ground temperature at the PACE borehole in Juvvasshøe at 0.4 m (upper) and 2.0 m (lower) depth. The shaded area indicates a period when no borehole data are available.**

At the rock glacier in Ivarsfjorden, a comparison between modelled and measured temperature is performed for average daily ground surface temperatures, using the mean of the measurements at 11 sites within the rock glacier as target for the comparison (Fig. 4). The best-fitting model configuration was found to be the *blocks with sediment* stratigraphy and a snowfall factor of 1.0, resulting in an RMSE of 1.3 °C and a bias of -0.4 °C. As in Juvvasshøe, the configuration used *drained* conditions, while differences with *undrained* conditions are small. Figure 4 also shows the significant spatial variability of ground surface temperatures, which is particularly large in winter. Also in periods, when the simulation results and the mean of the measurements visibly deviate, the simulations remain within the range of the measurements. While there are some deviations between the observations and simulation results at both Juvvasshøe and Ivarsfjorden, we conclude that the model setup, including the model forcing, can capture the general ground surface temperature regime at both sites which is a prerequisite for obtaining meaningful results from the *equilibrium* and *transient runs*.

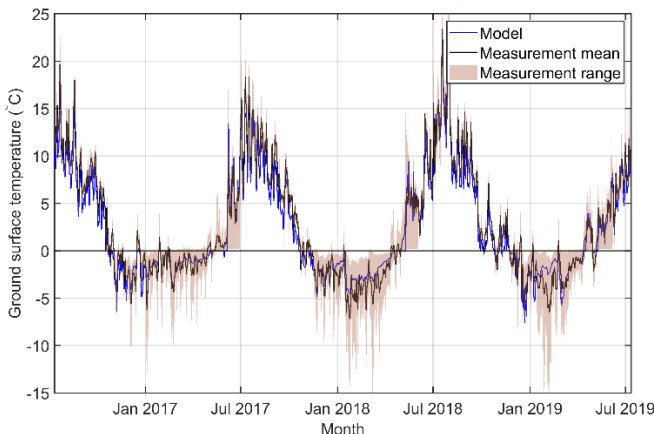

350

Figure 4: Daily modelled and measured ground surface temperatures in Ivarsfjorden from July 2016 to July 2019. The shaded area
indicates the minimum to maximum range of measured daily values from 11 loggers (based on Lilleøren et al., 2022), while the black
line represents the mean value of all loggers.

**4.2 Equilibrium ground temperatures and sensitivity to snow**

Figure 5 shows the average ground temperature at 2 m depth for the three stratigraphies, the *drained* and the *undrained*

scenario, and different snowfall factors at both sites. At both sites there is a clear pattern of lower temperatures in the *blocks*

*only, drained* scenario (solid blue line) compared to all five other scenarios. For snowfall factors of 0.75 and larger, the

difference in ground temperature between *blocks only, drained* and the other scenarios is in the range of 1.1 °C and 1.8 °C at

Juvvasshøe and in the range of 1.1 °C and 2.2 °C at Ivarsfjorden. This shows that the magnitude of the negative thermal

anomaly increases with a larger amount of snowfall. Results of the sensitivity study to porosity of the soil (Table S1 in the

Supplement) show that mean ground temperatures are within 0.4 °C between the highest and lowest porosity tested.

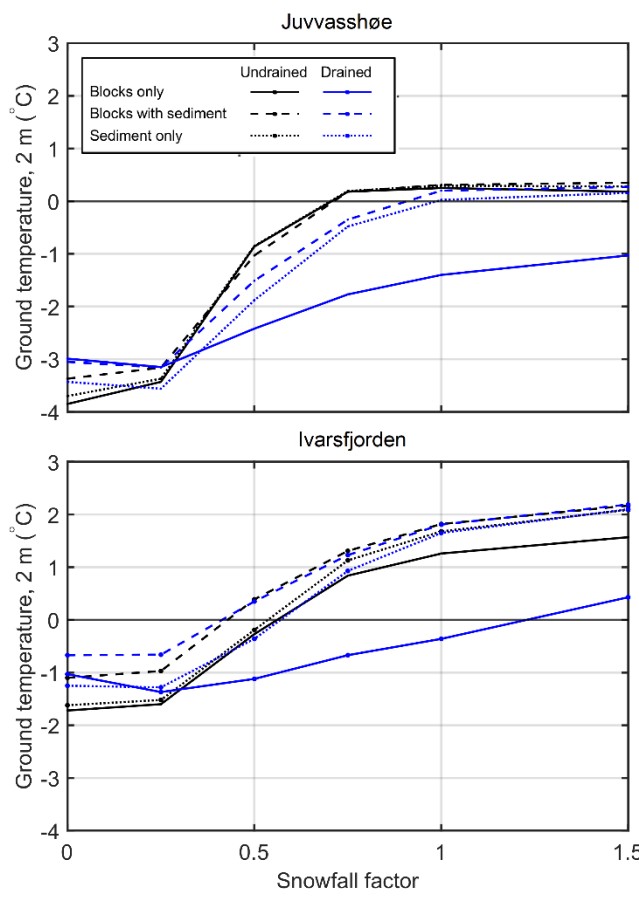

**Figure 5: Equilibrium ground temperature at 2 m depth for three idealized stratigraphies (Table 1) and different snowfall factors. Each data point represents one model run of one of the six scenarios at a certain snowfall factor.**

Annual maximum snow depths at a snowfall factor of 1.0 are between 1.5 m and 2.4 m at Juvvasshøe and between 0.4 m and 1.0 m at Ivarsfjorden. At Juvvasshøe, all three *undrained* scenarios feature positive ground temperatures at snowfall factors of 0.75 and above, which corresponds to permafrost-free conditions. Temperatures in the *blocks with sediment, drained* and *sediment only, drained* runs are positive for a snowfall factor of 1.0 and above. The ground temperature in the *blocks only, drained* runs remains below -1.0 °C for all snowfall scenarios. A similar pattern is seen in Ivarsfjorden, although a snowfall factor of 1.5 results in positive temperatures for the scenario *blocks only, drained* which is clear evidence of the overall warmer ground temperatures. Temperatures for the *blocks with sediment* stratigraphy are positive for snowfall factors exceeding 0.5, and exceeding 0.75 for the other scenarios (with the exception of the *blocks only, drained* scenario, see above). For snowfall factors above 0.25, ground temperature at 2 m depth increase with snow depth as a result of increased insulation of the ground during winter. However, the increase from a snowfall factor of 0 to 0.25 leads to a slight cooling for the *drained* scenarios as opposed to a slight warming in the *undrained* scenarios. The reason for this cooling is likely the higher winter albedo of the

completely snow-free ground (for snowfall factor zero), which outweighs the insulating of the shallow snow cover for snowfall
factor 0.25.
Figure 6 shows simulated temperatures for one year at the ground surface and 2 m depth for drained conditions for
both *blocks only* and the *blocks with sediment* scenarios for Juvvasshøe (snowfall factor 0.75). While ground surface
temperatures are largely similar during the snow-free summer season, they decrease much faster in fall for the *blocks only*
compared to the *blocks with sediment* scenario, for which the slow refreezing of the active layer leads to a prolonged warming
of the ground surface from below. In the *blocks only* scenario, on the other hand, the active layer contains only little water, so
that refreezing occurs within only a short time period. The rapid cooling in the *blocks only* scenario is also visible within the
permafrost table at 2 m depth, resulting in lower winter temperatures compared to the *blocks with sediment* curve and thus
explaining the simulated differences in mean ground temperature.

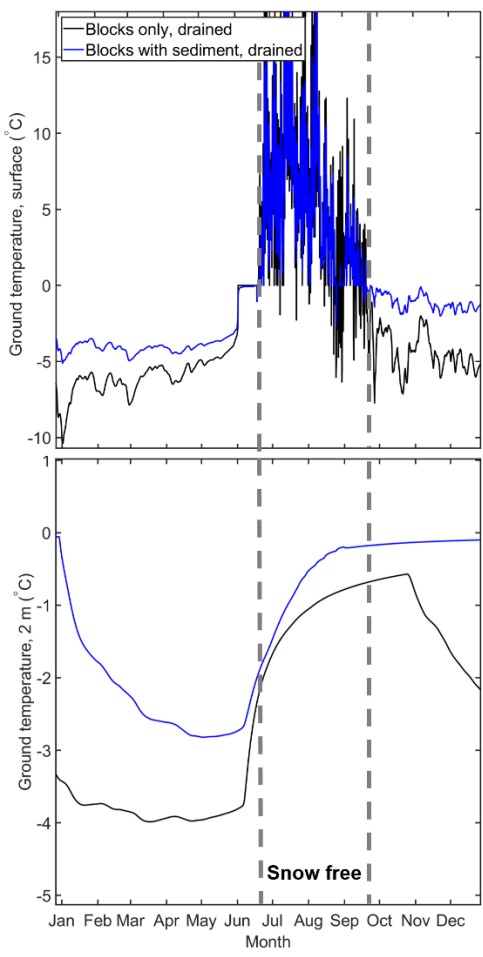


**Figure 6: Modelled ground temperature at 0.05 m (top) and 2 m (bottom) depth for the *blocks only, drained* and *blocks with sediment,***
***drained* scenarios during a year of an equilibrium run at Juvvasshøe for *sf = 0.75*. The snow-free summer season is highlighted. Note**
**that the upper plot is truncated at 17 °C, maximum summer temperatures are 26 °C in both scenarios.**

Figure 7 shows the corresponding snow cover and volumetric ground ice content in the upper meter of the ground for
the *blocks only, drained* scenario. A largely stable ground ice table forms already at a depth of about 0.5 m, while the active
layer is almost free of ground ice in winter, corresponding to the low water contents for thawed condition, enabling rapid
refreezing and thus strong cooling during winter. During and after snow melt, meltwater infiltrates in the blocky layer and
refreezes at the then very cold ice table, resulting in the formation of new ground ice which slowly melts during the course of
summer. The slight increase of the ground ice table in early winter is due to refreezing of residual water above the ice table
from rain and snow melt events in October which has not fully drained before refreezing.

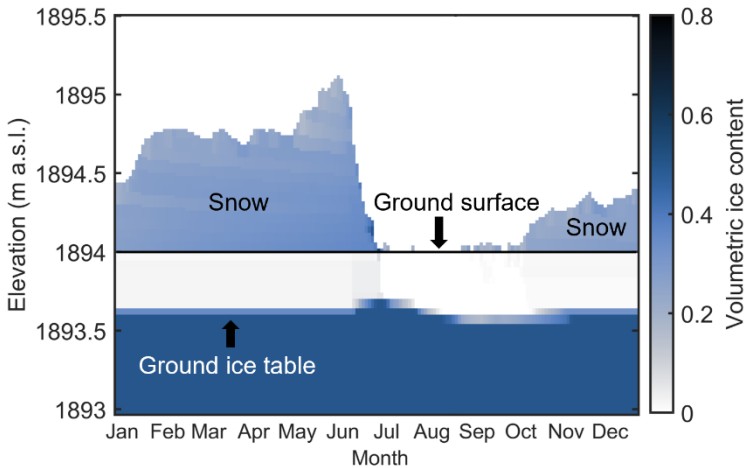


**Figure 7: Modelled volumetric ground ice content in the upper 1 m of the ground (below 1894 m a.s.l.) and the snow cover (above**
**1894 m a.s.l.) for the *blocks only, drained* scenario, during one year of an equilibrium run at Juvvasshøe for *sf* = 0.75. Note the rise**
**of the ground ice table, here defined as the uppermost cell where ground ice persists for two or more years, in June after infiltrated**
**snow melt water refreezes.**
**4.3 Transient response to climate warming**
The ERA5 reanalysis dataset allows us to simulate the evolution of the ground thermal regime and ground ice content from
1951 to 2019, during which mean air temperatures increased from -4.5 °C (1951-1960) to -3.8 °C (2010-2019) for Juvvasshøe
and from 0.5 °C (1951-1960) to 1.2 °C (2010-2019) at Ivarsfjorden. Figure 8 shows the ground ice content for different
scenarios in Ivarsfjorden. In all simulations, a stable ice table and permafrost conditions form during the spin up period (using
model forcing for the cold period 1962-1971, Table 2), with volumetric ice contents of 0.5 (*blocks only, sediment only*) and of
0.25 (*blocks with sediment*) according to the applied stratigraphy (Table 1). In the period 1951 to 2019, ground ice content
evolves as a response to the applied climate forcing, showing different responses of the ground ice table. In the *blocks only,*
*drained* scenario, the ice table in the upper 5 m (so between the active layer and the bedrock) does not lower by a significant
amount (0.1 m lowering), while the ice table lowers by 1.2 m in the *blocks only, undrained* scenario. The ice table in the *blocks*
*with sediment* stratigraphy disappears by 1985 and 1975 in the *undrained* and *drained* scenarios, respectively. Finally, the
*sediment only* simulations show an intermediate effect where the ice table has lowered by 1.7 m and 1.6 m for *undrained* and
*drained* conditions respectively by 2019. The complete degradation in the *blocks with sediment* runs compared to partial
degradation in all other scenarios (except *blocks only, drained*) is not unexpected since this stratigraphy has a 25% porosity
(and thus ice content), compared to 50% in the others. We conclude that the ground stratigraphy and drainage conditions
strongly control the response of the ground towards warming, with full degradation near-surface permafrost in both *blocks*
*with sediment* runs, partial degradation in the *blocks only, undrained* run and in both *sediment only* runs and finally continued
stable permafrost conditions in the *blocks only, drained* simulation. At the Juvvasshøe site, the ice table remain stable in all
simulations, but a slight lowering occurs in the *blocks with sediment* scenarios.

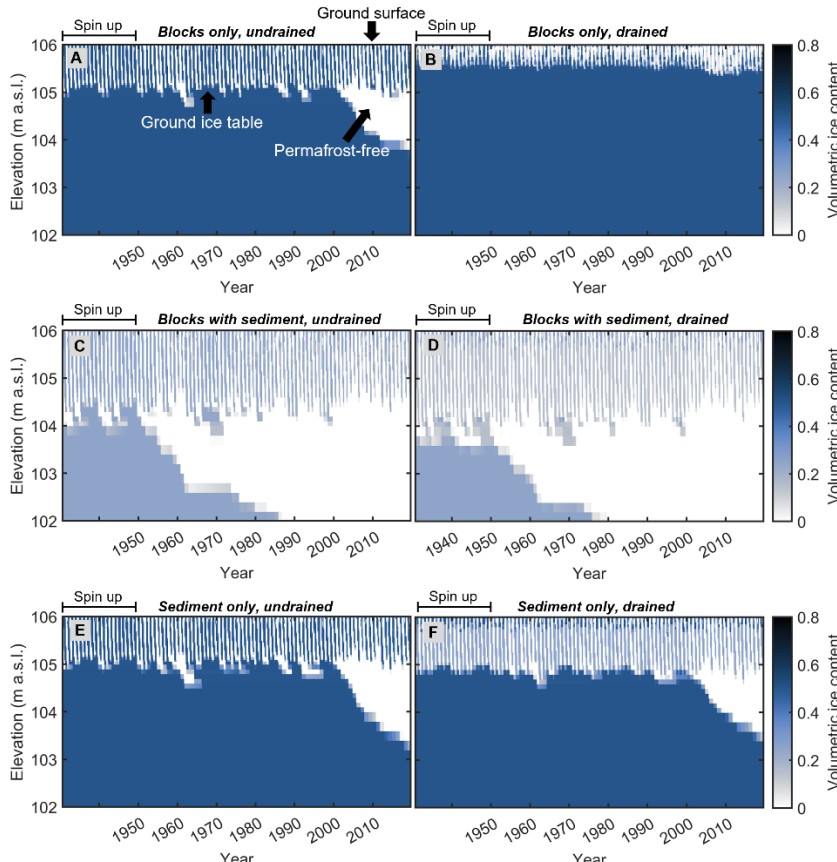

**Figure 8: Modelled volumetric ground ice content at Ivarsfjorden between 1951 and 2019 for the idealized stratigraphies in**
***undrained*** **and** ***drained*** **conditions and** *sf* **= 1.0. Only the subsurface domain is shown, with the ground surface elevation at 106 m**
**a.s.l.. In the active layer, ice contents increase and decrease annually, corresponding to the active layer refreezing and thawing. The**
**ground ice table is defined as the uppermost cell where ground ice persists for more than two years and follows the permafrost table.**
Figure 9 shows the change in temperatures at 5 m depth for the *drained* scenarios, which at the Ivarsfjorden rock glacier
(snowfall factor 1.0) correspond to a full (*blocks with sediment*) and partial (*sediment only*) lowering of the ice table, as well
as a relatively stable (*blocks only*) ice table. The *blocks only* simulation shows an increase from -0.6 °C to -0.2 °C between the
1951–1960 and 2010–2019 means, not being strongly influenced by latent heat effects due to the relative stable ice table. The
*sediment only* case experiences only minimal warming, as it is strongly influenced by the ongoing ground ice melt which
confines ground temperatures to close to 0 °C. Finally, the complete disappearance of ground ice in *blocks with sediment* run
coincided with in a warming to positive temperatures, from 0.0 °C to 0.6 °C. At Juvvasshøe, permafrost degradation and thus
strong ground ice melt does not occur for any of the scenarios for snowfall factor 0.25, and ground temperatures only increased
by 0.2 °C (*blocks only*) to 0.4 °C (*blocks with sediment* and *sediment only*) between the 1951–1960 and 2010–2019 means
(Fig. 8). The results of the transient runs indicate that the subsurface stratigraphy and drainage conditions strongly affect the
timing of permafrost degradation in blocky terrain. While ground ice melt controls the warming rates at Ivarsfjorden, only
small differences in warming rates are simulated for the still stable permafrost in Juvvasshøe. However, we emphasize that the
simulations at Juvvasshøe were performed for a shallow snow cover (*sf* = 0.25) for which differences in modeled ground
temperatures are small (Fig. 5).

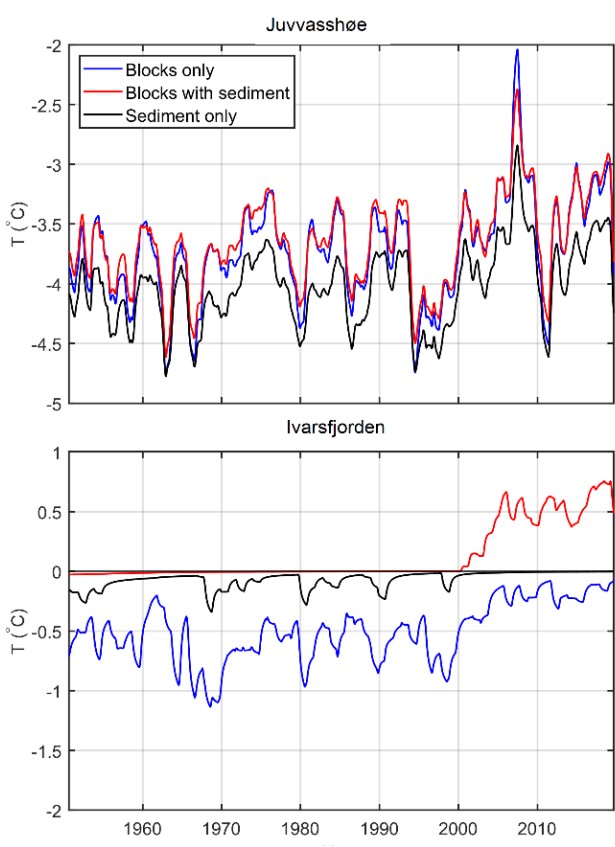


Figure 9: Ground temperature at 5 m depth for the idealized stratigraphies under *drained* conditions; *sf* = 1.0 for Ivarsfjorden and *sf* = 0.25 for Juvvasshøe.

## 5. Discussion

### 5.1 Limitations of the model setup

In this study, CryoGrid has been applied at two permafrost sites in Norwegian mountain environments. At both sites, we set up validation runs to benchmark the performance of model system against measurements of ground (Juvvasshøe) and ground surface (Ivarsfjorden) temperatures. At Juvvasshøe, the model can largely reproduce the annual cycle of measured ground temperatures at the PACE borehole, when the snowfall is reduced to account for the generally shallow snow cover at the site. At Ivarsfjorden, simulations with full snowfall yielded a similar performance for the ground surface temperature, approximately reproducing the mean of measurements at 11 sites. A statistical evaluation at both sites indicated a cold bias of the model of approximately -0.5 °C which we considered acceptable, considering the spatial variability of the ground thermal regime at both sites (see Gisnås et al., 2014 for Juvvasshøe). At Ivarsfjorden, the transient simulations are in broad agreement with observations at the rock glacier which indicate that permafrost has been present in the recent past (Lilleøren et al., 2022). Permafrost conditions are simulated for all stratigraphies during model spin using the cold period 1962-1971 for which temperatures are closest to Little Ice Age conditions when the rock glacier was likely active.

Within the model setup, in particular the exact ground stratigraphy and other poorly constrained parameters, such as the albedo, give rise to uncertainties. While the real porosity of the ground is unknown, sensitivity tests show a maximum of 0.4 °C differences in simulated ground temperatures between the highest and lowest porosity values tested (Table S1 in the Supplement). Only at Juvvasshøe, the stratigraphy has been described from the borehole (Isaksen et al., 2003), while no thorough evaluation of the subsurface stratigraphy is available for Ivarsfjorden. Lilleøren et al. (2022) described the site as a complex creeping system with inhomogeneous subsurface properties. Most of the rock glacier surface is described as 'relict' (Lilleøren et al., 2022) with sand and gravel in between blocks. For these 'relict' areas, the simulations for the *blocks with sediment* stratigraphy, in which near-surface permafrost fully or partially degrades, could indeed represent the thermal state adequately. This is supported by the validation run with the *blocks with sediment* stratigraphy which yielded a good fit with ground surface temperature measurements at sites largely located on this 'relict' surface (Lilleøren et al., 2022). Two areas are described as 'fresh' which could indicate lateral movements due to the presence of ground ice. These contain larger blocks and could thus be better described by the *blocks only* stratigraphy for which permafrost and ground ice still persist at the end of the simulations. However, also in these 'fresh' areas, the amount of finer sediment is unclear, in particular in deeper layers. In our simulations, we have only considered a single, homogeneous layer in the uppermost 5 m in order to compare the thermal regime and ground ice dynamics for idealized stratigraphies. In reality, ground stratigraphies in blocky terrain can feature aspects of all scenarios, for example a blocky layer with air-filled voids on top, followed by blocks filled with sediments and a sediment only layer in the bottom. For the cooling effect described in this study, it is critical that the blocky top layer is deep

enough so that a ground ice table from which water can drain can form within. Therefore, it is likely that also shallower blocky
layers with air-filled voids can lead to lower ground temperatures, depending on the climatic conditions which determine the
depth of the ground ice table.
We emphasize that a consistent model setup was selected for all scenarios, so that uncertainties caused by other parts
of the model system influence them all in a similar, consistent way. In particular, none of the convective processes summarized
by Harris and Pedersen (1998) that cause a negative thermal anomaly in blocky terrain are considered in the model setup. The
same applies to the effect of rocks protruding into and through the snow cover as was described by Juliussen and Humlum
(2008) which could potentially be included in CryoGrid by laterally coupled simulations (e.g. Zweigel et al., 2021) with snow
redistribution between tiles representing blocks of different heights. Considering air convection in future simulations (as e.g.
in Wicky and Hauck, 2017) should become a priority for model development as this is likely to interact with the ground ice
mass balance for the blocky drained scenario and could thus exacerbate the thermal anomaly.
Further uncertainties are related to the model forcing data. The ERA5 reanalysis data is a global product with coarse
horizontal resolution, so that the TopoSCALE downscaling routine (Fiddes and Gruber 2014) is applied to obtain more
representative meteorological forcing. Nonetheless, as mentioned in Fiddes and Gruber (2014) and Fiddes et al. (2019; 2022),
there are limitations to this scheme, in particular the primitive downscaling scheme for precipitation, which only interpolates
between ERA5 grid points and thus misses the effects of local orography. The same is true for the effects of local cloud build-
up around slopes and mountains, which affects the radiation budget. While these uncertainties could affect the comparison of
model results to field measurements (Sect. 4.1), the model forcing data can certainly capture the regional-scale climate
characteristics of the two study sites, e.g. the significant differences in MAAT between the two sites. The thermal anomaly of
the *blocks only, drained* scenario consistently occurs for both sites and thus over a significant range of climate conditions, so
that the effect is likely robust despite the uncertainties in the model forcing data. The same is true for the uncertainty caused
by the Crocus-based snow scheme (Vionnet et al., 2012; Zweigel et al., 2021). In this study, we have performed a sensitivity
study with respect to the amount of snow (by modifying the snowfall factor, Sect. 4.2), but simply scaling snowfall cannot
represent the true time evolution of the snow cover due to wind redistribution (e.g. Liston & Sturm, 1998; Martin et al., 2019),
possibly resulting in differences between observed and simulated temperatures. Nevertheless, it seems unlikely that the exact
time dynamics of snow ablation and/or deposition events strongly affects the dependence of the thermal anomaly in the *blocks*
*only, drained* scenario on overall winter snow depths. We therefore conclude that the significant negative thermal anomaly for
the *blocks only, drained* scenario is likely robust in the light of the model uncertainty.
**5.2 The effect of the ground ice dynamics on ground temperatures**
Despite the uncertainties of the model setup, our results show a clear negative thermal anomaly for the *blocks only, drained*
scenario. If the winter snow depth is sufficiently high, a surface cover of coarse blocks with air-filled voids (i.e. high porosity
and low water holding capacity) results in 2 m ground temperatures 1 to 2 °C lower than for the other stratigraphies. In the
Ivarsfjorden simulations, the *blocks only, drained* scenario is the only one where near-surface permafrost conditions persist

even today, while near-surface permafrost degrades for the *blocks with sediment* and *sediment only* scenarios. This is accompanied by a strong thermal offset, with a mean ground surface temperature of more than 2 °C for the *blocks only, drained* scenario, while the mean ground temperatures at 2 m were below 0 °C. Interestingly, the temperature anomaly appears largely constant over time in the transient simulations, except for periods when permafrost disappears in one of the scenarios and confines ground temperatures to 0 °C, which delays further ground warming. For lower snow depths, the temperature anomaly becomes smaller and eventually vanishes for the (largely irrelevant) case of permanently snow-free conditions.

The negative temperature anomaly largely accumulates during fall and winter (Fig. 6). The active layer contains very little water in the *blocks only, drained* scenario. Dry soils have a lower thermal conductivity compared to wet soils, but the lack of latent heat release allows for rapid refreezing during fall which enables fast cooling of the deeper soil layers and thus leads to overall lower winter temperatures. In spring, this "cold content" (i.e. sensible heat) of the ground is partly transformed into the build-up of new ground ice (i.e. latent heat, Fig. 7) which only melts slowly during summer due to the insulation of the overlying blocky layer. This timing of the ground ice formation is strongly different from all other scenarios, for which ground ice mostly forms in fall/early winter due to refreezing of the water contained in the active layer (e.g. Hinkel et al., 2001). A somewhat similar effect has been described for peat plateaus in northern Norway where simulations yielded 2 °C lower temperatures for well-drained peat compared to water-saturated peat (Martin et al., 2019). This refreezing of the active layer can take several months and is further delayed if a significant snow cover forms during this period, which leads to overall higher winter temperatures in the permafrost due to the insulation (Zhang, 2005). It is exactly for these "high-snow situations" (corresponding to higher snowfall factors in our sensitivity analyses) that the temperature anomaly of the *blocks only, drained* scenario is largest. Our results for example suggest that permafrost can occur for blocky ground on slopes around Juvvasshøe, even if the winter snow cover exceeds 2 m thickness.

We note that the thermal anomaly caused by the ground ice dynamics in blocky ground is not related to convective processes (Harris and Pedersen, 1998) or the effect of blocks protruding through the snow cover (Juliussen and Humlum, 2008; Gruber Hoelzle, 2008). The simulated temperature anomaly is similar to the 1.3–2.0 °C lower temperatures that Juliussen and Humlum (2008) found in blockfields compared to till and bedrock in Central-eastern Norway. While a complete process model for blocky ground and rock glaciers will certainly have to take air convection and the interplay between surface blocks and the snow cover into account, it is encouraging that the relatively simple model approach presented in this work offers prospects to improve our estimates of permafrost occurrence in mountain environments.

In a first-order approach, thermal anomalies can be translated into elevation differences by assuming a temperature lapse rate, so that the impacts on the lower altitudinal limit of permafrost can be estimated. For a lapse rate of 0.5 °C per 100 m (e.g. Farbrot et al., 2011), the lower limit of permafrost in drained, blocky deposits in Norway would be 300 to 400 m lower compared to "normal" permafrost represented by the other scenarios. For the Ivarsfjorden site, these numbers compare favorably to the Scandinavian permafrost map (Gisnås et al., 2017) which shows a lower discontinuous permafrost limit in Finnmark at around 400 m a.s.l., approximately 300 m above the rock glacier.

## 5.3 Implications for future work

In this study, we show that modelling the full subsurface water and ice balance in well-drained blocky deposits with air-filled voids leads to significantly lower ground temperatures in permafrost environments. In modelling studies on the distribution of permafrost, blocky ground usually is not accounted for, or the water balance is not simulated at this level of detail. For the Northern hemisphere permafrost map (Obu et al., 2019), a coarse landcover classification was used in mountain areas which did not represent blocky terrain. To produce the map, a semi-empirical equilibrium TTOP model was used in which the thermal anomaly of blocky deposits likely could account for by adjusting the $r_k$ parameter which accounts for the thermal offset of the ground. A more sophisticated modelling approach as presented in this work could be used to train the more simple TTOP model across a range of climate conditions. Permafrost mapping with transient models (e.g. Jafarov et al., 2012) often uses fixed ground stratigraphies, in which the sum of water and ice contents does not change for a given layer. An example is the transient permafrost map for southern Norway (Westermann et al., 2013) which featured a dedicated stratigraphic class for blocky deposits, with a dry upper layer followed by an ice-saturated layer below, very similar to the stratigraphy used for the *blocks only, drained* scenario in this study. However, both layers have a fixed thickness and the sum of water and ice contents is constant, so that the temporal evolution of ground ice dynamics cannot be captured. If the seasonal thaw extends in the ice-rich layer, a pool of meltwater forms which cannot drain and hence strongly delays refreezing in fall, potentially resulting in the degradation of permafrost. In our simulations with full water/ice dynamics, the ice table instead varies over time, both seasonally and over longer periods in response to the climatic forcing. Such changes in the ground ice table have for example been observed at the Schilthorn site in the European Alps where the ground ice table was significantly lowered during a hot summer and did not regrow in the following years although permafrost conditions persisted (Hilbich et al., 2008). As this observation site is located on a slope, it is clear that such observed ground ice dynamics can only be reproduced if lateral drainage of meltwater is taken into account. To improve transient modelling of mountain permafrost distribution, CryoGrid in the configuration used in this study could be adapted for individual grid cells, especially by adjusting the strength of the lateral drainage (i.e. the distance to seepage face) depending on the local slope. In flat areas and depressions, water would then pool up as for the *undrained* cases, while both melt- and rainwater would drain in sloping terrain as in the *undrained* cases, with corresponding changes to the ground thermal regime and permafrost distribution. Furthermore, our study suggests that the presence of fine sediments in the voids between blocks can strongly alter the ground temperature compared to blocky terrain with air-filled voids. For spatially distributed mapping, these two cases would have to be distinguished as separate stratigraphic classes and maps of their spatial extent must be available. Especially the latter is expected to be significant challenge, as the surfaces likely appear similar for remote sensors, so that detailed field mapping may be required.

The model approach in this study also offers significant potential to study ground ice derived runoff from blocky deposits and rock glaciers. While the Norwegian study sites are both located in wet climate settings with ample water supply, rock glaciers in more arid regions can be important sources of water (e.g. Croce and Milana, 2002). The global ratio of rock glacier to glacier water volume equivalent is currently increasing as both systems react differently to a changing climate (Jones

et al., 2019). Therefore, simulations of ground ice volumes and seasonal runoff characteristics in both the present and future climates can be a valuable tool for the assessment of water resources. Furthermore, rock glaciers are sensitive to climate change (Haeberli et al., 2010) and recent studies have linked rock glacier acceleration to increasing air and ground temperatures (e.g. Käab et al., 2007; Hartl et al., 2016; Eriksen et al., 2018; Thibert and Bodin, 2022). Our model approach is likely able to simulate the seasonal ground ice mass balance at different points and elevations of a rock glacier which could be ingested in a flow model for rock glaciers (e.g. Monnier and Kinnard, 2016). Finally, permafrost degradation and ground ice loss can also play an important role for slope stability in mountain permafrost environments (e.g. Gruber and Haeberli, 2007; Saemundsson et al., 2018; Nelson et al., 2001). Simulations of ground ice table changes, as well as the occurrence of strong melt events with corresponding production of meltwater, could eventually improve assessments of the stability and hazard potential of permafrost-underlain slopes (e.g. Mamot et al., 2021).

## 6. Conclusions

In this study, we used the CryoGrid permafrost model to simulate the effect of blocky terrain on the ground thermal regime and ground ice dynamics at two Norwegian mountain permafrost sites (Juvvasshøe and Ivarsfjorden rock glacier). In particular, we investigated the effect of subsurface drainage, as typical on slopes, for three idealized stratigraphies, named *blocks only*, *blocks with sediment* and *sediment only*. From this study, the following conclusions can be drawn:

- Markedly lower ground temperatures are found in well drained, coarse blocky deposits with air-filled voids (*blocks only, drained* scenario) compared to other scenarios which are either undrained or feature fine sediments. This negative thermal anomaly can exceed 2 °C and is mainly linked to differences in the freeze-thaw dynamics caused by the removal of meltwater and the build-up of new ground ice in spring. The largest anomalies occur in simulations with a thick winter snow cover as ground temperatures in well drained blocky deposits are less sensitive to insulation by snow than other soils. We emphasize that the model does not account for well-known factors, such as air convection and the effect of blocks protruding through the winter snow cover.

- For the *blocks only, drained* scenario, thermally stable permafrost can exist at the Ivarsfjorden rock glacier site (located near sea level), even for a mean annual ground surface temperature of 2.0–2.5 °C. At Juvvasshøe in the southern Norwegian mountains, permafrost is simulated even for a very thick winter snow cover in the *blocks only, drained* scenario, while all other scenarios in this case feature permafrost-free conditions.

- Transient simulations since 1951 at the Ivarsfjorden rock glacier show a completely or partially degraded ground ice table for all scenarios, except the *blocks only, drained* scenario. This result is explained by the overall lower ground temperatures in this scenario, while the simulated warming rates are generally similar for all scenarios, except for periods when strong ground ice melt occurs.

This study suggests that including subsurface water and ice dynamics can drive simulations of mountain permafrost dynamics
towards reality, which can for example improve estimates of the lower altitudinal limit of permafrost in blocky terrain. In
addition to permafrost distribution mapping, the presented model approach could be used to simulate the seasonal and multi-
annual evolution of the ground ice table, in addition to ground-ice derived runoff. It therefore represents a further step to a
better understanding and model representation of the permafrost processes in mountain environments.

*Code and data availability.* The CryoGrid source code and model setup files are available
https://doi.org/10.5281/zenodo.6563651 (Renette, 2022). Field measurements at Juvvasshøe are from Etzelmüller et al. (2020).
Field measurements at Ivarsfjorden are from Lilleøren et al. (2022).

*Author contribution.* CR performed the model simulations, retrieved forcing data, wrote the draft manuscript and created all
figures. SW helped design the study, developed the model and provided ideas throughout the entire study. KA developed code
for retrieving and downscaling forcing data, assisted with this process and wrote text regarding the forcing data. KL and KI
provided field measurements, site descriptions and photos. RBZ and JA developed parts of the model. All authors contributed
with text and suggestions.

*Competing interests*. The authors declare that they have no conflict of interest.

*Acknowledgements*. This work was supported by ESA Permafrost_CCI (https://climate.esa.int/en/projects/permafrost/),
Permafrost4Life (Research Council of Norway, grant no. 301639), and Nunataryuk (EU grant agreement no. 773421), as well
as the Department of Geosciences, University of Oslo.

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
