# Peer review of "Simulating the effect of subsurface drainage on the thermal regime and ground ice in blocky terrain, Norway"

_Earth Surface Dynamics, 2022_

## Referee Comment (RC2)

**Simulating the effect of subsurface drainage on the thermal regime and ground ice in blocky terrain, Norway**

Cas Renette[1], Kristoffer Aalstad[1], Juditha Aga[1], Robin Benjamin Zweigel[1,2], Bernd Etzelmüller[1], Karianne Staalesen Lilleøren[1], Ketil Isaksen[3], Sebastian Westermann[1]

[1]Department of Geosciences, University of Oslo, Oslo, Norway
[2]Centre for Biogeochemistry of the Anthropocene, UiO, Oslo
[3]Norwegian Meteorological Institute, Oslo, Norway

*Correspondence to*: Cas Renette (cas.renette@gmail.com)

**Abstract.** Ground temperatures in coarse, blocky deposits such as mountain blockfields and rock glaciers have long been observed to be lower in comparison with other (sub)surface material. One of the reasons for this negative temperature anomaly is the lower soil moisture content in blocky terrain, which decreases the duration of the zero curtain in autumn. Here we used the CryoGrid community model to simulate the effect of drainage in blocky terrain permafrost at two sites in Norway. The model setup features a surface energy balance, heat conduction and advection, as well as a bucket water scheme with adjustable lateral drainage. We used three idealized subsurface stratigraphies, denoted *blocks only*, *blocks with sediment* and *sediment only* and are either *drained* or *undrained* of water, resulting in six scenarios'. The main difference between the three stratigraphies is their ability to retain water against drainage: while the *blocks only* stratigraphy can only hold small amounts of water, much more water is retained within the sediment phase of the two other stratigraphies, which critically modifies the freeze-thaw behaviour.

The simulation results show markedly lower ground temperatures in the *blocks only, drained* scenario compared to other scenarios, with negative thermal anomaly of up to 1.8–2.2 °C. For this scenario, the model can in particular simulate the time evolution of ground ice, with build-up during and after snow melt and spring and gradual lowering of the ice table in the course of the summer season. We simulate stable permafrost conditions at the location of a rock glacier in northern Norway with a mean annual ground surface temperature of 2.0–2.5 °C in the *blocks only, drained* simulations. Finally, transient simulations at the rock glacier site showed a complete or partial lowering of the ground ice table since 1951 for all simulations except the *blocks only, drained* run.

        The interplay between the subsurface water/ice balance and ground freezing/thawing driven by heat conduction can at least partly explain the occurrence of permafrost in coarse blocky terrain below the assumed elevational limit of permafrost. It is thus important to consider this effect in future efforts on permafrost distribution mapping in mountainous areas.

Furthermore, an accurate prediction of the evolution of the ground ice table in a future climate can have implications for slope stability, as well as water resources in arid environments.

**Summary of Comments on esurf-2022-39.pdf**

**Page: 1**

Number: 1 Author: Reviewer    Subject: Comment on Text    Date: 9/7/22, 9:32:49 PM
effect of drainage on what? soil thermal regime? please explain

Number: 2 Author: Reviewer    Subject: Comment on Text    Date: 9/7/22, 9:32:49 PM
please explain here what type of model domain was considered? Is it a 2D generic hillslope or observed 2D/3D topography?

Number: 3 Author: Reviewer    Subject: Inserted Text    Date: 9/7/22, 9:32:49 PM
denoted by

Number: 4 Author: Reviewer    Subject: Comment on Text    Date: 9/7/22, 9:32:49 PM
please clarify what does drained/undrained mean here. saturated or unsaturated conditions?

Number: 5 Author: Reviewer    Subject: Comment on Text    Date: 9/7/22, 9:32:49 PM
remove quotes

Number: 6 Author: Reviewer    Subject: Comment on Text    Date: 9/7/22, 9:32:49 PM
I assume it means, there is more permafrost in the blocks only, drain case. How much in terms of percentage as compared to other two cases?

Number: 7 Author: Reviewer    Subject: Comment on Text    Date: 9/7/22, 9:32:49 PM
would be nice to provide a rough number here, like what is that limit you are refering to?

Number: 8 Author: Reviewer    Subject: Comment on Text    Date: 9/7/22, 9:32:49 PM
this effect? this is unclear. does it refer to the role of heat conduction, which is 'mostly' included in permafrost simulation, or does it say 'consider simulating permafrost below that assumed elevational limit?'

[Figure]

**1 Introduction**

Permafrost is defined as ground that remains at or below 0 °C for two or more consecutive years (Van Everdingen, 1998) and is a common feature in
mountain environments. In discontinuous mountain permafrost terrain, the lowest-lying active permafrost landforms are frequently found in coarse, blocky terrain (Harris and Pedersen, 1998). In particular, rock glaciers are frequently found below the assumed elevational limit of permafrost (Lilleøren and Etzelmüller 2011).

       The occurrence of a negative temperature anomaly in coarse, blocky deposits has long been recognized, e.g. in central eastern Norway by Liestøl (1966). Harris and Pedersen (1998) found a negative temperature anomaly of 4 to 7 °C in blocky terrain relative to adjacent mineral sediment in mountains in Canada and China. They summarized four hypotheses that explain these anomalies: (a) The Balch effect; (b) chimney effect; (c) evaporation of water and sublimation of ice in the summer and (d) continuous air exchange with the atmosphere when no continuous winter snow cover is present.

       Juliussen and Humlum (2008) showed that block fields in the Norwegian mountains featured a negative temperature anomaly of 1.3 to 2.0 °C, which was mainly attributed to rocks protruding into and through the snow cover which leads to a higher effective thermal conductivity of the snow cover. Gruber and Hoezle (2008) presented a simple model for the conductive effect of blocks protruding through the snow cover which showed that the mean annual ground temperature is reduced as a result of a lower thermal conductivity of a blocky layer.

       Additionally, a lower soil moisture content in permeable blocky debris decreases the duration of the zero-curtain in autumn since less latent heat is liberated compared to soil with higher soil moisture content (Juliussen and Humlum 2008). In spring, the opposite effect is observed when percolating meltwater refreezes at the bottom of the blocky surface layer and confines temperatures at the ice interface to 0 °C (e.g. Juliussen and Humlum, 2008; Hanson and Hoelzle, 2004; Humlum, 1997).

       Rock glaciers play an important role in the hydrological cycle, especially in arid regions like the Andes, where in some areas more water is stored in rock glaciers than in glaciers (Jones et al., 2019; Azócar and Brenning, 2010). The open debris structure can act as a trap for snow and a rock glacier can store a significant quantity of ice or liquid water. Rock glaciers studied in Argentina are an important water resource as they release water mainly during periods of drought (Croce and Milana 2002). Ground ice melt as a response to climate warming threatens this water source. Additionally, melting of ground ice can lead to slope instability (e.g. Gruber and Haeberli, 2007; Saemundsson et al. 2018; Nelson et al., 2001) and damage to infrastructure (e.g. Arenson et al., 2009).

       In southern Norway, permafrost underlies large parts of areas above 1500 m.a.s.l.. The permafrost elevation limit decreases from above 1600 m.a.s.l. in the west to about 1100 m.a.s.l. in the eastern, more continental areas (Etzelmüller et al., 2003). In northern Norway, the limit is around 800–1000 m.a.s.l in the west and decreases towards the east. A first inventory of Norwegian rock glaciers based on aerial imagery was published in 2011 (Lilleøren and Etzelmüller, 2011), suggesting that found active permafrost landforms occur above 400 m.a.s.l.. The density of rock glaciers in Norway is lower than in other mountain permafrost areas which was attributed to a lack of bedrock competence and debris availability as well as to the

**Number: 1 Author: Reviewer    Subject: Comment on Text    Date: 9/7/22, 9:32:49 PM**
be specific here. not all mountain environments have permafrost

**Number: 2 Author: Reviewer    Subject: Comment on Text    Date: 9/7/22, 9:32:49 PM**
for a smooth flow, at least mention different types of permafrost here,  such as continuous, discontinuous, sporadic.

**Number: 3 Author: Reviewer    Subject: Comment on Text    Date: 9/7/22, 9:32:49 PM**
again, what is that assumed elevation limit

**Number: 4 Author: Reviewer    Subject: Comment on Text    Date: 9/7/22, 9:32:49 PM**
not clear if this is authors' analysis of Juliussen and Humlum work or they are just refering to their work.

**Number: 5 Author: Reviewer    Subject: Cross-Out    Date: 9/7/22, 9:32:49 PM**

[Figure]

relative lack of steep topography above the permafrost limit. Recently, Lilleøren et al. (2022) described rock glaciers near sea level in the area of Hopsfjorden, northern Norway, which feature a limited ice body and are in transition from active to relict. Additionally, Nesje et al. (2022) presented new evidence for active rock glaciers in southern Norway well below the assumed permafrost limit. Warming of Norwegian mountain permafrost (Etzelmüller et al., 2020) is expected to continue in the 21st century (Hipp et al., 2012), resulting in further degradation of these ice bodies and an upward shift of the lower permafrost limit. Hipp et al. (2012) also mentioned the need to address the effect of snow cover and surface material on how ground temperatures respond to climate forcing.

Land surface models that can represent permafrost are vital tools to investigate the sensitivity to climate change and complex environmental conditions. Since permafrost presence is often not visible at the surface, numerical modelling based on process understanding is often used to estimate the permafrost distribution (Harris et al. 2009). One-dimensional heat flow models have been used in studies to investigate the effect of climate change on permafrost (e.g. Etzelmüller et al., 2011; Hipp et al., 2012) or to model specific processes in mountain permafrost (e.g. Gruber and Hoezle, 2008). However, many such models do not include a transient representation of the subsurface water and ground ice balance (e.g. Etzelmüller et al., 2011; Hipp et al., 2012; Westermann et al., 2013).

The CryoGrid community model (Westermann et al., 2022) is a simulation toolbox that can calculate ground temperatures and volumetric water as well as ice content in permafrost environments. It largely builds on the well-established CryoGrid 3 model (e.g. Westermann et al., 2016; Martin et al., 2019) and accommodates a broad range of applications. In the following, the CryoGrid community model is referred to as "CryoGrid" for simplicity.

In this study, we present simulations of ground temperatures and ice content for different idealized subsurface stratigraphies and drainage regimes using CryoGrid, applied at two Norwegian permafrost sites: a blockfield site in southern

Norway and a rock glacier in northern Norway. The aim is to contribute to an improved process understanding of the coupled subsurface energy and water/ice balance, focusing on the observed negative thermal anomaly in coarse, blocky deposits. In particular, we present simulations how the ground ice mass balance in blocky terrain can affect ground temperatures and the occurrence of permafrost.

**2 Study sites**

**2.1 Juvvasshøe, southern Norway**

Juvvasshøe (61°40 N, 08°22 E, 1894 m.a.s.l.) (Fig. 1) is a site located in the southern Norwegian mountains, Jotunheimen well above the tree line. A 129 m deep borehole was drilled in August 1999 in the PACE (Permafrost and Climate in Europe) project (Harris et al., 2001). Continuous data streams from this PACE borehole are available with the exception of a gap between 21 December 2011 to 24 April 2014. The site is located in an extensive block field on a mountain plateau with sparse vegetation cover. The bedrock (crystalline rocks, Farbrot et al., 2011) is located at approximately 5 m depth, the first meter consists of large stones and boulders and the ground below mainly consists of cobbles (Isaksen et al., 2003). Between 2000

Number: 1 Author: Reviewer    Subject: Comment on Text    Date: 9/7/22, 9:32:49 PM
please provide some specific examples of applications

Number: 2 Author: Reviewer    Subject: Comment on Text    Date: 9/7/22, 9:32:49 PM
rephrase

Earth **Surface**
Dynamics
Discussions

[Figure]

and 2004, Isaksen et al. (2007) measured a mean annual air temperature (MAAT) at 2 m height of -3.3 °C. The mean ground temperature (MGT) at 2.5 m below the surface during this period was -2.5 °C. The mean precipitation was estimated to be between 800 and 1000 mm $a^{-1}$. The site is extremely wind-exposed, resulting in a low snow thickness due to wind drift. Hipp et al. (2012) described a snow depth of less than 20 cm, while the snow thickness in surrounding, lower-lying and less exposed sites can be up to 140 cm. Isaksen et al. (2007) measured the difference between the MAGST and MAAT (surface offset) at exposed and less exposed sites in this area. At sites with a significant snow cover, the surface offset was more than 2 °C, while at exposed (including Juvvasshøe) sites this offset is generally below 1 °C. The permafrost thickness at the PACE borehole was estimated to be approximately 380 m (Isaksen et al., 2001), with the lower permafrost limited at ca. 1450 m.a.s.l. (Farbrot et al., 2011). A weak zero curtain effect suggests a low water content in the active layer (Isaksen et al., 2007). A warming of 0.2 °C per decade and 0.7 °C per decade in surface air temperature and ground surface temperature, respectively, occurred between 2000 and 2019 (Etzelmüller et al., 2020).

[Figure]

**Figure 1: Location of the two sites in Norway (© Norwegian Mapping Authority). (1) blockfield at Juvvasshøe (1894 m.a.s.l.), (2)**
**rock glacier at Ivarsfjorden (60–160 m.a.s.l.).**

Number: 1 Author: Reviewer    Subject: Comment on Text    Date: 9/7/22, 9:32:49 PM
mean annual ground?

Number: 2 Author: Reviewer    Subject: Comment on Text    Date: 9/7/22, 9:32:49 PM
mean annual? I see the unit here is mm/yr (rate),  but general practice is to provide "mean" in mm (amount)

Number: 3 Author: Reviewer    Subject: Cross-Out    Date: 9/7/22, 9:32:49 PM

Number: 4 Author: Reviewer    Subject: Comment on Text    Date: 9/7/22, 9:32:49 PM
I guess, Mean annual ground surface temperature? this is never defined before. please define it before using it.

Number: 5 Author: Reviewer    Subject: Comment on Text    Date: 9/7/22, 9:32:49 PM
and how deep is the permafrost table? i.e., depth of permafrost from the surface at the exposed sites and/or sites with significant snow cover. If you have this information available it would be good to add.

[Figure]

Earth **Surface**
Dynamics
Discussions

**2.2 Ivarsfjorden rock glacier, northern Norway**

Ivarsfjorden is a small fjord arm of the larger Hopsfjorden, located on the Nordkinn peninsula in the Troms and Finnmark county in northern Norway (Fig. 1). Deglaciated around 14–15 cal kyr BP (Romundset et al., 2011), the peninsula is dominated by flat mountain plateaus of exposed bedrock, *in situ* weathered material and coarse grained till (Lilleøren et al., 2022), which
feature steep slopes towards the sea. The coastal areas of Finnmark have a wet maritime climate, with mean annual precipitation around 1000 mm (Saloranta, 2012). Lilleøren et al. (2022) describe a MAAT of 1.6 °C between 2010 and 2019 in the area of the rock glacier. The rock glacier of interest lies in a southwest-northeast trending valley that extends from the fjord and has an elevation extent of roughly 60 to 160 m.a.s.l.. The mountain at its east (443 m.a.s.l.) serves as the source area with rockfall debris and coarse talus slopes being common. The bedrock in Ivarsfjorden consists of sandstones and phyllites (NGU, 2008).
Sandstones often generate coarse, bouldery material, which is favorable for the formation of rock glaciers (Haeberli et al. 2006). The rock glacier in Ivarsfjorden is northwest facing and has previously been interpreted as relict (Lilleøren and Etzelmüller, 2011), but a detailed analysis showed that a limited ice core might still be present (Lilleøren et al., 2022). A negative MAAT around 100 to 150 years ago is an indication that rock glaciers in this area were active at the end of the Little Ice Age (LIA). Refraction Seismic Tomography (RST) surveys indicate a porous air-filled stratigraphy such as blocky talus
deposits. While observed MAGSTs between 2015 and 2020 are all positive, negative surface temperatures during summer have been observed by a thermal camera at the front slope of the rock glacier. This is likely an indication of the chimney effect and thus of connected voids that support air flow.

**3. Methods**

**3.1 The CryoGrid community model**

CryoGrid is a simulation toolbox for ground thermal simulations that can be applied to a wide range of applications in the terrestrial cryosphere thanks to its modular structure (Westermann et al., 2022). It has mainly been applied in permafrost environments, using heat conduction to simulate ground temperatures transiently. To represent the energy and water cycle of a one-dimensional ground column in the best possible way, CryoGrid allows the user to select different processes representations/parameterizations for different vertical domains. As an example, a fully transient water balance can be used in
the near-surface region, while a constant water(ice) content is used in deeper layers. Likewise, different process representations for the seasonal snow cover can be chosen. To define the properties of the subsurface material, a stratigraphy of volumetric mineral, organic, water and ice content must be provided (Westermann et al., 2022).

In the model setup used for this study, the lower boundary condition is provided by a constant geothermal heat flux of 0.05 Wm$^{-2}$ which is a reasonable value for Norway used in previous modeling studies (Westermann et al., 2013). The upper
boundary results from solving the full surface energy balance, including both radiative and turbulent heat fluxes, which gives rise to a ground heat flux. In order to solve the surface energy balance, atmospheric forcing data are required (chapter 3.4). A

Number: 1 Author: Reviewer     Subject: Comment on Text     Date: 9/7/22, 9:32:49 PM
it needs to be defined, if it has not been defined already

Number: 2 Author: Reviewer     Subject: Comment on Text     Date: 9/7/22, 9:32:49 PM
how is this depth picked for constant water/ice content layer? and how is this depth related to the damping depth of the surface thermal signal?

Number: 3 Author: Reviewer     Subject: Comment on Text     Date: 9/7/22, 9:32:49 PM
this also needs some explanation. Like what type of snow processes? aging, compaction, density variations etc.? It would be hard for someone not familiar with CryoGrid to follow it.

[Figure]

scheme with heat conduction, following Fourier's law for heat conduction, and heat advection is used for heat transfer and temperature calculation in the subsurface (Westermann et al., 2022).

For soil hydrology, a gravity driven bucket scheme is used. Rainfall is taken from the forcing data and added to the uppermost cell of the subsurface. The surface energy balance calculations determine how the soil moisture is affected by evaporation. Transpiration plays no part in this study as no vegetated surfaces are involved. Water that is in excess of the field capacity infiltrates downwards until either the water table or a frozen cell is reached. The water table forms if excess water is available and cells are saturated from the bottom upwards. If all grid cells are saturated, excess water is considered as surface runoff.

The freezing characteristic of water in the subsurface can be set to follow a freeze curve depending on the soil type, following Painter and Karra (2014) or set as free water (water changes state at 0 °C, Westermann et al., 2022) for subsurface material with large pores/voids, such as blocky terrain.

Studies that used a previous model version of CryoGrid (e.g. Westermann et al., 2013; Westermann et al., 2016; Langer et al., 2016) assumed constant water/ice contents. This is a major limitation, as varying soil moisture contents strongly affect the ground thermal regime (e.g. Martin et al., 2019). We use a one-dimensional model setup and simulate lateral drainage out of the model domain by assuming a seepage face at atmospheric pressure. First, the elevation of the water table is calculated, after which a lateral water flux removes water in grid cells that are below this water table, following Eq. (1):

$$F_i^{lat} = -\mathrm{K}_H \frac{z_{\mathrm{wt}} - z_{\mathrm{i}}}{d^{lat}}, \tag{1}$$

where $F_i^{lat}$ is the lateral water flux out of grid cell $i$, $\mathrm{K}_H$ is the hydraulic conductivity, $z_{\mathrm{wt}}$ is the height of the water table, $z_{\mathrm{i}}$ is the height of grid cell $i$ and $d^{lat}$ is the lateral distance to the seepage face. We use the parameter $d^{lat}$ to control the strength of the drainage, where low values result in a well-drained column and high values result in a poorly or completely undrained column. A $d^{lat}$ value of $10^4$ m is used for *undrained* cases, which emulates conditions at a mostly flat surface. In reality this corresponds to no drainage, resulting in a one-dimensional case for the water balance, where only surface water is removed. For the *drained* cases, a $d^{lat}$ value of 1 m is used, which emulates well-drained conditions at a slope.

The snow model used in this study was introduced by Zweigel et al. (2021) and is based on the CROCUS snow scheme (Vionnet et al. 2012). Snowfall is added on the surface with density and grain properties derived from atmospheric forcing data and then undergoes transient evolution of snow grains and density. The snow albedo undergoes a transient evolution in this snow scheme. The physical effect of wind drift on the snowpack is included in the module as well. Energy and mass transfer in the snowpack includes heat conduction, percolation of rainfall and percolation of meltwater (Westermann et al., 2022). The amount of snow can be adjusted by a so-called *snowfall factor, sf,* with which the snowfall from the model forcing is multiplied. With this, it is possible to phenomenologically represent redistribution of snow by wind (as in Martin et al., 2019) and to account for potential biases in the snowfall forcing. In this study, we conduct a sensitivity analysis towards the winter snow cover by selecting different values for *sf*.

Number: 1 Author: Reviewer     Subject: Comment on Text     Date: 9/7/22, 9:32:49 PM
how do you compute the water table locaton? explain

Number: 2 Author: Reviewer     Subject: Comment on Text     Date: 9/7/22, 9:32:49 PM
above this water table?

[revised manuscript text omitted]

Number: 1 Author: Reviewer    Subject: Comment on Text    Date: 9/7/22, 9:32:49 PM

Seems to me that Figure 2 does not support this statement, as at depth 4 m, the model fails to follow the observed trend particularly in the spring shoulder season for all years. Is it due to bottom boundary condition or snow model or thermal conductivites?

How the authors ensured that the simulations won't be affected by the bottom boundary condition if a 5 m domain is used?

[Figure]

[Figure]

[Figure]

**Figure 2: Modelled and measured ground temperature at the PACE borehole in Juvvasshøe at 0.4 m (upper) and 2.0 m (lower)**
**depth. The shaded area indicates a period when no borehole data are available.**

At the rock glacier in Ivarsfjorden, no borehole data are available. Therefore, a comparison between modelled and measured temperature is performed with data from a ground surface temperature logger network (Lilleøren et al., 2022). As this site is characterized by complex microtopography with likely strong variability of the snow cover (for which no observations are available), we focus on mean annual ground surface temperatures (MAGST) for individual years instead of a time-resolved comparison as for Juvvasshøe. The three years of data at 11 locations on and near the rock glacier are presented together with six model *validation runs* (Fig. 3). These consist of two idealized stratigraphies (Sect. 3.3), in addition to the blockfield stratigraphy (see Table 1, Westermann et al., 2013) each in a *drained* and *undrained* configuration with a snowfall factor 1.0. There is likely a variable snow depth across the different parts of the rock glacier, which possibly explains part of the differences between modelled and measured temperatures.

Measured MAGST are in the range of 1.1 °C and 4.1 °C, while modelled MAGST are confined in a narrower range between 2.0 °C and 2.7 °C. For all years combined, the mean of the measurements is 2.7 °C, while the mean of all model realizations is 2.2 °C. We emphasize that the simulations all use a single snowfall factor and identical surface conditions, which likely explains the smaller range of simulated temperatures. In reality, the loggers are distributed in an area where small scale spatial variations in topography, snow accumulation, ground stratigraphy and vegetation play a role. While there are some deviations between measurements and simulation results at both Juvvasshøe and Ivarsfjorden, we conclude that the

Number: 1 Author: Reviewer    Subject: Comment on Text    Date: 9/7/22, 9:32:49 PM

how well modeled MAGST matches the observed for the other site?

[Figure]

model setup (including the model forcing) 1an capture the general ground surface temperature regime at both sites which is a prerequisite for obtaining meaningful results from the *equilibrium* and *transient runs*.

[Figure]

**Figure 3: Modelled and measured MAGST in Ivarsfjorden during three years, from 13 July 2016 to 12 July 2019. The bars indicate**
**the 25th and 75th percentile of measured MAGST and the whiskers represent the maximum and minimum temperatures. The blue indicators show modelled MAGST during the same period for a selected range of ground stratigraphies at *sf* = 1.**

**4.2 Equilibrium ground temperatures and sensitivity to snow**

Annual maximum snow depths at a snowfall factor of 1.0 are between 1.5 m and 2.4 m at Juvvasshøe and between 0.4 m and
1.0 m at Ivarsfjorden. Figure 4 shows the equilibrium ground temperature at 2 m depth for the three stratigraphies at different snowfall factors at both sites. For each of the three stratigraphies there is the *drained* and the *undrained* scenario. At both sites there is a clear pattern of lower temperatures in the *blocks only, drained* scenario (solid blue line) compared to all 5 other scenarios. For snowfall factors of 0.75 and larger, the difference in ground temperature between *blocks only, drained* and the other scenarios is in the range of 1.1 °C and 1.8 °C at Juvvasshøe and in the range of 1.1 °C and 2.2 °C at Ivarsfjorden.

Number: 1 Author: Reviewer     Subject: Comment on Text     Date: 9/7/22, 9:32:49 PM

I found it confusing. Capturing general trends in the ground surface temperature is no guarantee that soil thermal state is represented accurately due to the complex and nonlinear nature of the subsurface.

[Figure]

Earth **Surface**
**Dynamics**
Discussions

[Figure]

Figure 4: Equilibrium ground temperature at 2 m depth for three idealized stratigraphies (Table 1) and different snowfall factors.

At Juvvasshøe, all three undrained scenarios feature positive ground temperatures at snowfall factors of 0.75 and above, which corresponds to permafrost-free conditions. Temperatures in the *blocks with sediment, drained* and *sediment only, drained* runs are positive for a snowfall factor of 1.0 and above. The ground temperature in the *blocks only, drained* runs
remains below -1.0 °C for all snowfall scenarios. A similar pattern is seen in Ivarsfjorden, although a snowfall factor of 1.5 results in positive temperatures for the scenario *blocks only, drained* which is clear evidence of the overall warmer ground temperatures. Temperatures for the *blocks with sediment* stratigraphy are positive for snowfall factors exceeding 0.5, and exceeding 0.75 for the other scenarios (with the exception of the *blocks only, drained* scenario, see above). For snowfall factors above 0.25, ground temperature at 2 m depth increase with snow depth as a result of increased insulation. However, the increase
from a snowfall factor of 0 to 0.25 leads to a slight cooling for the *drained* scenarios as opposed to a slight warming in the *undrained* scenarios. The reason for this cooling is likely the higher winter albedo of the completely snow-free ground (for snowfall factor zero), which outweighs the insulating of the shallow snow cover for snowfall factor 0.25.

Earth **Surface**
Dynamics
Discussions
EGU

Fig. 5 shows simulated temperatures for one year at the ground surface and 2 m depth for drained conditions for both *blocks only* and the *blocks with sediment* scenarios for Juvvasshøe (snowfall factor 0.75). While ground surface temperatures are largely similar during the
[Figure]
 snow-free summer season, they decrease much faster for the *blocks only* compared to the *blocks with sediment* scenario, where the slow refreezing of the active layer leads to a prolonged warming of the ground surface from below. In the *blocks only* scenario, on the other hand, the active layer contains only little water, so that refreezing occurs within only a short time period. The rapid cooling in the *blocks only* scenario is also visible within the permafrost table at 2 m depth, resulting in lower winter temperatures compared to the *blocks with sediment* curve and thus explaining the simulated differences in MAGT.

[Figure]

**Figure 5:** Modelled ground temperature at 0.05 m (top) and 2 m (bottom) depth for the *blocks only, drained* and *blocks with sediment, drained* scenarios during a year of an equilibrium run at Juvvasshøe. *Sf* = 0.75.

Number: 1 Author: Reviewer    Subject: Comment on Text    Date: 9/7/22, 9:32:49 PM
highlight the summer reason in Fig. 5

Number: 2 Author: Reviewer    Subject: Comment on Text    Date: 9/7/22, 9:32:49 PM
The upper plot seems truncated vertically. Either adjust it or mention it in the caption

[Figure]

Figure 6 shows the corresponding snow cover and volumetric ground ice content in the upper meter of the ground for the *blocks only, drained* scenario. A largely stable ground ice table forms already at a depth of about 0.5 m, while the active layer is almost free of ground ice in winter, corresponding to the low water content enabling rapid refreezing and thus strong cooling during winter. During and after snow melt, meltwater infiltrates in the blocky layer and refreezes at the then very cold ice table, resulting in the formation of new ground ice which slowly melts during the course of summer. The slight increase of the ground ice table in early winter is due to refreezing of residual water above the ice table from rain and snow melt events in October which has not fully drained before refreezing.

[Figure]

**Figure 6: Modelled volumetric ground ice content in the upper 1 m of the ground (<1894m) and the snow cover (>1894m) for the *blocks only, drained* scenario, during one year of an equilibrium run at Juvvasshøe. Note the rise of the ground ice table in June after**
**infiltrated snow melt water refreezes. *Sf* = 0.75.**

**4.3 Transient response of ground temperatures and ice content**

The ERA5 reanalysis dataset allows us to model the evolution of the ground thermal regime and ground ice content from 1951 to 2019. Figure 7 shows the ground ice content for the different scenarios in Ivarsfjorden. In all simulations, a stable ice table and permafrost conditions form during the spin up period, with volumetric ice contents of 0.5 (*blocks only, sediment only*) and of 0.25 (*blocks with sediment*) according to the applied stratigraphy (table1). In the period 1951 to 2019, ground ice contents evolves as a response to the applied climate forcing, showing different responses of the ground ice table. In the *blocks only, drained* scenario, the ice table does not lower by a significant amount, while the ice table lowers significantly in all other simulations. The perennial ice table in the upper 5 m of the *blocks with sediment* stratigraphy disappeared by 1985 and 1975 in the *undrained* and *drained* scenarios respectively. Finally, the *sediment only* simulations show an intermediate effect where the ice table has dropped to approximately half of its initial height by 2019. We conclude that the ground stratigraphy and drainage conditions strongly control the response of the ground towards warming, with full degradation near-surface permafrost in both *blocks with sediment* runs, partial degradation in the *blocks only, undrained* run and in both *sediment only*

Number: 1 Author: Reviewer    Subject: Comment on Text    Date: 9/7/22, 9:32:49 PM

This needs more details, dry soils have low thermal conductivity, so how can it enable rapid refreezing? There is always a competition between the soil thermal conductivity and heat capacity.

Number: 2 Author: Reviewer    Subject: Cross-Out    Date: 9/7/22, 9:32:49 PM

Number: 3 Author: Reviewer    Subject: Comment on Text    Date: 9/7/22, 9:32:49 PM

1. What is the surface elevation? highlight it on the plots.

2. I wonder how was this model initialized that the authors got low water content around 104 m and above/below the water content is very different and high?

3. Same for bottom right plot. what is causing the patchy low water content areas?

[Figure]

runs and finally continued stable permafrost conditions in the *blocks only, drained* simulation. At the Juvvasshøe site, the ice table persists in all simulations, but a slight lowering occurs in the *blocks with sediment* scenarios.

[Figure]

**Figure 7:** Modelled volumetric ground ice content at Ivarsfjorden for the idealized stratigraphies in *undrained* and *drained* conditions. The *sf* = 1.0 in all simulations.

The changes in ground temperature are also strongly dependent on the stratigraphy. Figure 8 shows the change in temperatures at 5 m depth for the *drained* scenarios, which at the Ivarsfjorden rock glacier (snowfall factor 1.0) correspond to a full (*blocks with sediment*) and partial (*sediment only*) lowering, as well as a relatively stable (*blocks only*) ice table. The *blocks only* simulation underwent an increase from -0.6 °C to -0.2 °C between the 1951–1960 and 2010–2019 means, not being

Number: 1 Author: Reviewer    Subject: Comment on Text    Date: 9/7/22, 9:32:49 PM
Provide more details in the caption and letter the subplots for easy referencing.

strongly influenced by latent heat effects due to the relative stable ice table. The *sediment only* case experienced minimal warming in this period at 5 m depth, strongly influenced by the ongoing ground ice melt which confines ground temperatures to close to 0 °C. Finally, the complete degradation of the ice table in *blocks with sediment* run coincided with in a warming to positive temperatures, from 0.0 °C to 0.6 °C. Ground temperatures at 5 m depth were lower for snowfall factor 0.5 and resulted in less ice melt, so that the warming between the 1951–1960 and 2010–2019 means were from -0.8 °C to -0.3 °C in the *blocks only* stratigraphy, from -0.5 °C to 0.0 °C in the *sediment only* stratigraphy and from -0.1 °C to 0.1 °C in the *blocks with sediment* stratigraphy.

[Figure]

**Figure 8: Ground temperature at 5 m depth for the idealized stratigraphies under *drained* conditions. *sf* = 1.0 for Ivarsfjorden and *sf* = 0.25 for Juvvasshøe.**

At Juvvasshøe, permafrost degradation and thus strong ground ice melt does not occur for any of the scenarios for snowfall factor 0.25 (Fig. 8). [1]evertheless, the warming rates differed to some degree between the scenarios. In the *blocks only* simulation, 5m ground temperatures increased from -3.7 °C to -3.5 °C between the 1951–1960 and 2010–2019 means, from -3.8 °C to -3.4 °C for the *blocks with sediment* run ranged and from -4.1 °C to -3.7 °C in the *sediment only* run. For snowfall factor 0.5, the change in temperature at 5 m depth in *drained* scenarios were overall higher, i.e. from -3.1 °C to -2.6

Number: 1 Author: Reviewer    Subject: Comment on Text    Date: 9/7/22, 9:32:49 PM
This needs to be explored, that how much of subsurface warming is due to stratigraphy, soil moisture, and how much the change in the air temperature caused over the period 1951-2019, while keeping the effects of snow isolated as well.

[revised manuscript text omitted]

---

## Author Comment (AC1)

**Response to Anonymous Referee #1**

We would like to thank the reviewer for the detailed and useful comments, which have helped to improve the quality and readability of our manuscript. In the following, we provide a reply to the points discussed by the reviewer as well as the changes in the manuscript.

The comments of the reviewer are written in **bold**, the extracts of the manuscript in *italics* with changes highlighted in *blue*.

**Major comments:**

**1 The idea of the model experiment – testing the impact of drainage on the ground thermal regime – is very interesting and the approach seems sound and straightforward. However, the model validation and the links to the two sites in Norway are weak. For one of the sites there are observed temperatures available to compare with the model results but these data are not used to validate the model results in a robust way. Instead, validation is only carried out visually and no objective statistics are used to evaluate the model results. For the other site, there are no data on ground temperatures but only observed ground surface temperatures. The validation and comparison to observations from the Norwegian sites thereby adds next to nothing to the modeling study. My suggestion would be to either remove these sites from the manuscript, to add some statistical analysis based on the available ground temperature data, and/or to find a rock glacier site with available ground temperature data which could be used for a real validation of model results.**

The two Norwegian field sites are of high importance to the study. They are used for validation with the available data sets and the comparison of observed and modelled data shows that the model can capture the main characteristics of the seasonal evolution of measured temperatures. We agree with the reviewer that temperature data from boreholes would be beneficial. However, as these do not exist for Norwegian rock glaciers, Ivarsfjorden is a well suited field site for this study, as it is well documented by a previous study (Lilleøren et al., 2022) and near-surface ground temperatures are recorded by several temperature loggers. Furthermore, the presence of the rock glacier landform itself, as well as geophysics investigations on ground ice presence, allows to constrain the permafrost evolution at least in a qualitative fashion (Lilleøren et al., 2022), and his can also be compared to the simulation results. We therefore decided not to remove the site from the study, but improve validation. To do so, we have followed the suggestion to include a statistical analysis of the model performance at both sites. Secondly, the model validation for the rock glacier site (Ivarsfjorden) has been redone in a much improved and more robust manner. Please find below the changes in our manuscript.

We added a more detailed description of the temperature loggers at Ivarsfjorden to explain the setup for model validation:

*Line 287-291 : At Ivarsfjorden, we considered 11 temperature loggers within the rock glacier outline (Fig. 1d in Lilleøren et al., 2022), of which all except for one are placed on the relict surface of the rock glacier (Fig. 2a in Lilleøren et al., 2022). On the relict surface, deposition of finer sediment in between blocks is more likely than on the active surface, due to the lack of movement. Here, the blocks with sediment stratigraphy is considered appropriate and used as starting point for the calibration.*

Temperatures resulting from the best-fitting model setup are now presented in a similar way as Juvvasshøe, showing the model can reproduce daily ground surface temperatures in a satisfactory manner.

[Figure]

*Line 349-352 : Figure 4: Daily modelled and measured ground surface temperatures in Ivarsfjorden from July 2016 to July 2019. The shaded area indicates the minimum to maximum range of measured daily values from 11 loggers (based on Lilleøren et al., 2022), while the black line represents the mean value of all loggers.*

At both of the sites, the bias and RMSE are calculated for daily averages.

*Line 291-293 : At both sites, the root-mean-square-error (RMSE) and bias are calculated in order to provide an objective measure of the model fit. At Juvvasshøe this was accomplished for daily values at 0.4 m and 2 m depth, while at Ivarsfjorden the mean daily ground surface temperature of the loggers within the rock glacier outline is used.*

*Line 328-330 : This configuration used drained conditions, although differences with undrained conditions are minimal for this stratigraphy. For daily temperatures at 0.4 m depth, the RMSE and bias are 2.1 °C and -0.6 °C, respectively, while they are 1.2 °C and -0.7 °C at 2 m depth.*

*Line 340-341 : The best-fitting model configuration was found to be the blocks with sediment stratigraphy and a snowfall factor of 1.0, resulting in an RMSE of 1.3 °C and a bias of -0.4 °C.*

**2 The methods section would benefit from restructuring and extensive editing, because it is rather confusing as currently written.**

We agree that the method chapter was not clearly enough structured and required restructuring and clarification. In the revised manuscript, we have completely restructured the Methods section for clarity.

Section "3.1 The CryoGrid community model" contains a general description of the used model, not covering any site specifics or parameter values. We removed extensive descriptions of CryoGrid capabilities which are not used in our model setup. This section is followed by "3.2 Downscaling of model forcing", a description of the model forcing data, which gives the reader information about the time period and time resolution of the model forcing. Section "3.3 Model setup" now covers the model setup and parameters, which were previously scattered throughout the chapter. As suggested by the reviewer, we created separate subchapters for the Validation runs (3.3.1), Equilibrium runs (3.3.2) and Transient runs (3.3.3). Specific improvements / clarifications in the Methods will be highlighted at the corresponding review comment.

**2.1 Model description. It would be helpful to start the model description with a presentation of the mesh used: is this a vertical column (1D) mesh? What is the thickness and cell size of the domain?**

**Some of this information is available in other parts of the methods section, but starting off by clarifying such basic facts would help the reader to envision the model setup.**

In the process of improving and clarifying the methods section, we followed the reviewer's recommendation and included a schematic in the beginning of the methods section, that shows the model domain with its depth, cell sizes and boundary conditions. In this manner, the reader is directly informed about how the model domain looks.

*Line 154-155: We use a one-dimensional model column with a domain depth of 100 m (as in e.g. Westermann et al., 2016; Schmidt et al., 2021) and grid cell sizes increasing with depth (Fig. 2).*

[Figure]

*Line 165-168: Figure 2: Schematic of the model grid, indicating cell sizes at different depths and upper and lower boundary conditions. As upper boundary condition, the surface energy balance (SEB) forced by near-surface meteorological data is used. The lower boundary condition is provided by a constant geothermal heat flux at 100 m depth.*

**2.2 Snow model. A list of processes included in the model is provided, but there is no explanation of how these are represented in the model. If these are explained in previous publications, please clearly refer to those for the specific processes listed. For example, "the physical effect of wind drift on the snowpack" is included, according to the text, but later it is stated that the redistribution of snow by wind is phenomenologically represented using the snowfall factor. So, what effects of wind on the snowpack are actually included? Does the snowfall factor allow for variable distribution of snow on the domain, or is there also a redistribution of snow over time?**

We agree with the reviewer that better explanations and referencing is needed regarding the implemented snow processes in the model. We therefore refer to Vionnet et al. (2012) for defining equations of the Crocus snow model, and to Zweigel et al. (2021) for implementation in CryoGrid). Furthermore, we improved the explanation of the snow model in the manuscript, especially clarifying the snow redistribution.

*Line 189-197:The snow model used in this study was introduced by Zweigel et al. (2021) and is based on the Crocus snow scheme (Vionnet et al. 2012) which accounts for snow microphysics and is designed to reproduce a realistic snow pack structure (see Vionnet et al. 2012 for defining equations; Zweigel et al. 2021 for implementation in CryoGrid). Snowfall is added with density and microphysical properties derived from model forcing data, in particular air temperature and wind speed. The snow density evolves due to compaction by the overburden pressure of overlying snow layers, as well as wind compaction and refreezing of melt- and rainwater (Vionnet et al., 2012). The amount of snowfall from the forcing data can be adjusted by a so-called snowfall factor, sf, with which the snowfall rate from the model forcing is multiplied. With this, the effects of wind-induced snow redistribution on ground temperatures can be*

*represented at least phenomenologically (Martin et al., 2019), using sf < 1 for areas with net snow ablation and sf > 1 for areas with net deposition.*

**2.3 The description of the different model runs (validation, equilibrium, and transient) is rather confusing and split up in different sub-sections of the methods description. I would suggest to clarify the purpose of the model runs and explain them in a separate sub-section. The validation runs are runs validated based on visual fit, which is not a very robust metric for model validation. It seems like the purpose of validation runs was to determine (calibrate?) the proper sediment stratigraphy for each site. But how were other parameter values determined? Table 1 presents some physical properties, but not e.g. Kh. Snowfall factor is for some reason tested in the equilibrium runs (why?). In general, the purpose of testing the snowfall factor sensitivity is rather unclear to me. The outcome of this test is not mentioned in the abstract, but the current text does not suggest that this is pure model calibration. L230 suggests that snowfall factors were also tested in validation runs, but this is not mentioned previously.**

As mentioned in the response to major comment 2, we followed the suggestion to split the different model runs in sub-sections and clarified their purpose. The manuscript includes now the sections "3.3.1 Validation runs", "3.3.2 Equilibrium runs" and "3.3.3 Transient runs" and the purpose of the model runs has been clarified. Please refer to the corresponding sections of the revised manuscript for all changes. Specific comments are addressed below.

Model validation

We clarified the goal of the model validation:

*Line 271-272: Validation runs are set up to show that the model can reproduce key characteristics of the thermal regime at the two sites in a satisfactory manner (based on available observations).*

We agree with the reviewer that a visual fit is not sufficient, therefore we included a statistical analysis of the model performance and significantly improved validation for the rock glacier site (for changes in the manuscript see response to major comment 1).

We now describe that snowfall factors are tested in the model validation:

*Line 282-283: Manual adjustment of the ground stratigraphy (porosity and thus mineral content) and snowfall factor are performed until a good fit with daily measurements is achieved.*

Equilibrium runs

It is important to test different snowfall factors, as the thermal regime is highly sensitive towards snow conditions and the snow cover shows a high spatial variability in Norwegian mountains. Therefore, we investigated the sensitivity of ground temperatures by running the equilibrium runs with a range of snowfall factors, to estimate the threshold snow amount for permafrost existence in different model scenarios. We added further explanations to the revised manuscript by including previous studies and highlighting the purpose of the model runs.

*Line 295-301: The goal of equilibrium runs is to investigate the sensitivity of the ground thermal regime towards ground properties and drainage conditions, using both the undrained and drained setup for the three idealized stratigraphies (Table 1), which result in six scenarios. As the heavily wind-affected snow cover is a key source of spatial variability in ground temperatures in the Norwegian mountains (Gisnås et al., 2014; Signåa et al., 2016), the model is run for a range of snowfall factors between 0.0 and 1.5 (Table 2) for each scenario. This analysis allows us to identify the magnitude of the thermal anomaly that the subsurface drainage induces at various amounts of snow, as estimate the threshold snow amount for permafrost existence in the six scenarios.*

The outcome of this sensitivity test is now also included in the Abstract and Conclusions

*Line 24-25: The thermal anomaly increases with larger amounts of snowfall, showing that well drained blocky deposits are less sensitive to snow insulation than other soils.*

*Line 589-591: The largest anomalies occur in simulations with a thick winter snow cover as ground temperatures in well drained blocky deposits are less sensitive to insulation by snow than other soils.*

Model setup

We have provided further details on the choice of the hydraulic conductivity and the parameter $d^{lat}$ and also present further sensitivity studies in a Supplement:

*Line 245-255: To investigate the effect of subsurface drainage ground temperatures and ground ice conditions, we distinguish undrained and drained scenarios by using two different values of $d^{lat}$ (Eq. 1) for in the idealized stratigraphies. A $d^{lat}$ value of $10^4$ m is used for undrained cases, which emulates conditions at a flat surface, resulting in a to a good approximation one-dimensional water balance, where only surface water is removed. For the drained cases, a $d^{lat}$ value of 1 m is used, which results in well-drained conditions which are typical in sloping terrain. For the saturated hydraulic conductivity $K_H$, a fixed value of $10^{-5}$ m s$^{-1}$ is used for all stratigraphies, although the true hydraulic conductivities almost certainly differ between stratigraphies. However, the key parameter controlling lateral water fluxes in Eq. 1 is in reality the "drainage timescale" $K_H/d^{lat}$ [s$^{-1}$], which is varied by four orders of magnitude between $K_H/d^{lat} = 10^{-5}$ s$^{-1}$ ($d^{lat}$ = 1 m, well-drained conditions) and $K_H/d^{lat} = 10^{-9}$ s$^{-1}$ ($d^{lat}$ = $10^4$ m undrained conditions). As the study setup is designed to analyse these two "confining cases", it is sufficient to only vary $d^{lat}$ and leave $K_H$ constant for simplicity. Further sensitivity tests for $d^{lat}$ and $K_H$ are provided in the Supplement.*

**2.4 The forcing data used for spinup and simulation runs are also not presented in a clear manner. The spinup and initialization procedure and data vary between the sites and the runs and the motivation for the choice of spinup periods is not always clear. Perhaps a table presenting the basic details of each run would be more helpful. Also not clear in the current text: Is forcing data downloaded and downscaled for the two sites (e.i., two separate series of data)? What is the time resolution (daily? monthly?) of the data used for simulations and for validation of simulation results?**

We restructured the section "3.2 Downscaling of forcing data" and made distinct changes to the explanations, so that the procedure is now presented in a clearer manner. Specific comments are addressed below.

We present a table that gives an overview about the simulation types and includes information about the model forcing.

*Line 265-268: Table 2: Overview of basic model settings for the different simulations types. A spin-up of subsurface temperatures is achieved by repeated simulations for the spin-up period (until a stable temperature profile is reached), before the actual model run for the simulation period is conducted. "Idealized" stratigraphy and drainage refers to three subsurface stratigraphies (Table 1) combined with two types of drainage conditions. See Sect. 3.3.1 and Sect. 3.3.3 for details.*

| Simulation type | Site | Spin-up period | Simulation period | Stratigraphy and drainage | Snowfall factor |
|---|---|---|---|---|---|
| **Validation** | Juvvasshøe | 1951-2010 | 2010-2019 | Best-fit | 0.25 |
| | Ivarsfjorden | 1951-2016 | 2016-2019 | Best-fit | 1 |
| **Equilibrium** | Juvvasshøe | 2000-2010 | 2000-2010 | Idealized | 0.0, 0.25, 0.5, 0.75, 1.0, 1.5 |
| | Ivarsfjorden | 1962-1971 | 1962-1971 | Idealized | 0.0, 0.25, 0.5, 0.75, 1.0, 1.5 |
| **Transient** | Juvvasshøe | 1962-1971 | 1951-2019 | Idealized | 0.25 |
| | Ivarsfjorden | 1962-1971 | 1951-2019 | Idealized | 1 |

A separate ERA5 dataset is downloaded from Copernicus for both of the sites, using the nearest grid cells to the corresponding location. For both sites, the corresponding dataset is then downscaled with TopoScale. The time resolution of the downloaded ERA5 forcing data is three hours. This has been clarified in the text.

*Line 204-207: ERA5 output is provided as interpolated point values on a regular latitude-longitude grid at a resolution of 0.25° at an hourly frequency, both at the surface level, corresponding to Earth's surface as represented in the reanalysis, and at 37 pressure levels in the atmosphere from 1000 to 1 hPa. We considered data for the reanalysis period from 1951 to 2019 at three-hourly resolution.*

**Minor comments:**

**1 L155-164: Seepage face is at atmospheric pressure, but does that mean that the seepage face is located at Zwt? Could that sentence just be removed and eq. 1 explains all? Where are Kh values from? Eq. 1 mirrors Darcy's law but values seem made up, and dz is based on the grid z. Explain the rationale behind this model and why it was chosen. (alternatively, if Kh and dlat are unknown, these two parameters could have been replaced by one single parameter.)**

We agree that the rationale and the description of the seepage face was not clear. The entire paragraph was reformulated for clarity

*Line 173-185: We use a one-dimensional model setup, but simulate lateral drainage of water by introducing a seepage face, i.e. a lateral boundary conditions for water fluxes representing flow between saturated grid cells of the model domain and a stream channel (or the atmosphere) to which the water can freely flow out from the subsurface (e.g. Scudeler et al., 2017). Using the elevation of the water table, $z_{wt}$ (computed as the elevation of the uppermost saturated grid cell), lateral water fluxes $F_i^{lat}$ are derived for all saturated unfrozen grid cells i below the water table (i.e. at elevations $z_i < z_{wt}$ ) as*

$$F_i^{lat} = -K_H \frac{z_{wt} - z_i}{d^{lat}} \, ,$$

*where $K_H$ is the saturated hydraulic conductivity, $d^{lat}$ is the lateral distance to the seepage face and the flux is determined by the difference between the hydrostatic potential (proportional to $z_{wt}$) of the water column and the gravitational potential of free water at the elevation of each cell (proportional to $z_i$). Note that Eq. (1) is an approximation for small changes of the water table and small outflow fluxes for which the potential in the saturated zone can be approximated by the hydrostatic potential. The parameter $d^{lat}$ is used to control the strength of the drainage, with small distances resulting in a well-drained column, while high values lead to suppressed drainage.*

The concerns of the reviewer regarding the $K_H$ values are addressed in major comment 2.3

**2 L282: But table 1 does not include any tested case with a porosity of 0.2!! Either the methods section does not completely describe what was done in the validation runs or there is a typo here?**

As part of the improved methods chapter, we stated that the ground stratigraphy (porosity) is manually adjusted to find a good fit between measurements and simulated ground temperatures. The best fit was then used for model validation. We made the following changes to the text.

*Line 282-283: Manual adjustment of the ground stratigraphy (porosity and thus mineral content) and snowfall factor are performed until a good fit with daily measurements is achieved.*

**3 L296-312: The model results suggest that the differences in model setups have very little impact on MAGST and that overall there seems to be a slight cold bias in the model. What then does this validation add to our understanding, if the purpose of the study is to investigate the impact of drainage on ground temperatures and ground ice?**

The comparison to near-surface ground temperatures is clear evidence that the model can capture key processes, in particular the insulating effect of the snow cover, in a satisfactory way. We argue that is a prerequisite for being able to perform a meaningful sensitivity analysis for the effect of subsurface stratigraphies and drainage conditions. For the Ivarsfjorden site, the simulated ground surface temperatures indeed do not show a large spread between the different scenarios, but the good agreement with measurements shows that we capture the key processes relevant for the near-surface temperatures (i.e. snow cover, surface energy balance, etc.) Secondly, we show that (with the correct near-surface temperatures) it is possible to explain permafrost presence until today in the *blocks only, drained* scenario which fits e.g. to observations of cold air exiting the rock glacier (Lilleøren et al., 2022). Without the comparison to near-surface temperatures, one could simply select a low value for the snowfall factor which

would likely also explain permafrost presence, but such simulations would not provide agreement for near-surface ground temperatures. Sect. 5.1 now contains an in-depth discussion on this issue:

At Ivarsfjorden, simulations with full snowfall yielded a similar performance for the ground surface temperature, approximately reproducing the mean of measurements at 11 sites. A statistical evaluation at both sites indicated a cold bias of the model of approximately -0.5 °C which we considered acceptable, considering the spatial variability of the ground thermal regime at both sites (see Gisnås et al. 2014 for Juvasshøe). At Ivarsfjorden, the transient simulations are in broad agreement with observations at the rock glacier which indicate that permafrost has been present in the recent past (Lilleøren et al., 2022). Permafrost conditions are simulated for all stratigraphies during model spin using the cold period 1962-1971 for which temperatures are closest to Little Ice Age conditions when the rock glacier was likely active.

*Line 447-465: At Ivarsfjorden, simulations with full snowfall yielded a similar performance for the ground surface temperature, approximately reproducing the mean of measurements at 11 sites. A statistical evaluation at both sites indicated a cold bias of the model of approximately -0.5 °C which we considered acceptable, considering the spatial variability of the ground thermal regime at both sites (see Gisnås et al. 2014 for Juvasshøe). At Ivarsfjorden, the transient simulations are in broad agreement with observations at the rock glacier which indicate that permafrost has been present in the recent past (Lilleøren et al., 2022). Permafrost conditions are simulated for all stratigraphies during model spin using the cold period 1962-1971 for which temperatures are closest to Little Ice Age conditions when the rock glacier was likely active.*

*Within the model setup, in particular the exact ground stratigraphy and other poorly constrained parameters, such as the albedo, give rise to uncertainties. While the real porosity of the ground is unknown, sensitivity tests show a maximum of 0.4 °C differences in simulated ground temperatures between the highest and lowest porosity values tested (Supplement). Only at Juvvasshøe, the stratigraphy has been described from the borehole (Isaksen et al., 2003), while no thorough evaluation of the subsurface stratigraphy is available for Ivarsfjorden. Lilleøren et al. (2022) described the site as a complex creeping system with inhomogeneous subsurface properties. Most of the rock glacier surface is described as 'relict' (Lilleøren et al., 2022) with sand and gravel in between blocks. For these 'relict' areas, the simulations for the blocks with sediment stratigraphy, in which near-surface permafrost fully or partially degrades, could indeed represent the thermal state adequately. This is supported by the validation run with the blocks with sediment stratigraphy which yielded a good fit with ground surface temperature measurements at sites largely located on this 'relict' surface (Lilleøren et al., 2022). Two areas are described as 'fresh' which could indicate lateral movements due to the presence of ground ice. These contain larger blocks and could thus be better described by the blocks only stratigraphy for which permafrost and ground ice still persist at the end of the simulations.*

Please also refer to Major comments 1 and 2.

**4 Section 4.1: This section would greatly benefit from some objective measure of model fit instead of just some conclusions from visual inspection of plotted temperature curves.**

We agree with the reviewer and included a statistical analysis based on the available temperature data at both of the sites (see major comment 1).

**5 Section 4.2: It is not surprising that snow insulates the ground from winter cooling, generally leading to higher ground temperatures. More relevant than a comparison of snowfall factors, would be to check if observed snow depths and ground temperatures are recreated with the forcing data used here. When reading this section, I get curious about how much snow is generally observed at these two sites and does the forcing data capture the range of observed snow, or is a snowfall factor scaling needed to "correct" the forcing data**

It is true that the insulating effect of snow on ground temperatures is well known. However, as the snow conditions show a high spatial variability in Norwegian mountains and measurements have shown that

permafrost presence is often associated with a low snow cover, it is important to investigate the sensitivity of ground temperatures towards snowfall factors. Consequently, we consider the testing of snowfall factors to be relevant for this study. See as well the response to major comment 2.3.

The observed snow depth cannot be recreated directly with the forcing data, as wind redistribution plays an important role, especially at Juvvasshøe. Therefore, a snowfall factor was applied and fitted for model validation. This is described in the results of the original manuscript:

*Line 321-324: The snowfall factor for this model setup is 0.25, i.e. incoming snowfall is reduced 75% in order to capture the effect of snow ablation due to wind drift. This resulted in mean annual maximum snow depths of 34 cm, in broad agreement with observations from the site (Iskasen et al., 2003) and earlier modeling studies at the site (Westermann et al., 2013).*

We understand that a constant snowfall factor cannot represent the temporal evolution of the snow cover. This is discussed in section "5.1 Limitations of the model setup".

*Line 492-498: In this study, we have performed a sensitivity study with respect to the amount of snow (by modifying the snowfall factor, Sect. 4.2), but simply scaling snowfall cannot represent the true time evolution of the snow cover due to wind redistribution (e.g. Liston & Sturm, 1998; Martin et al., 2019), possibly resulting in differences between observed and simulated temperatures. Nevertheless, it seems unlikely that the exact time dynamics of snow ablation and/or deposition events strongly affects the dependence of the thermal anomaly in the blocks only, drained scenario on overall winter snow depths. We therefore conclude that the significant negative thermal anomaly for the blocks only, drained scenario is likely robust in the light of the model uncertainty.*

---

## Author Comment (AC2)

**Response to Anonymous Referee #2**

We would like to thank the reviewer for the detailed and useful comments, which have helped to improve the quality and readability of our manuscript. In the following, we provide a reply to the points discussed by the reviewer as well as the changes in the manuscript.

The comments of the reviewer are written in **bold**, the extracts of the manuscript in *italics* with changes highlighted in *blue*.

**Major comments:**

**1. Describe in more detail the rationale for each scenario.**

We follow the suggestion of the reviewer and describe the rationale of the model scenarios in more detail. This includes (1) a restructuring of the methods section with a more thorough explanation of the simulation types (3.3.1 Validation runs, 3.3.2 Equilibrium runs, 3.3.3 Transient runs). (2) Furthermore, we explained our choice of ground stratigraphies after each description of *blocks only, blocks with sediment* and *sediment only*:

*Line 220-237: The blocks only stratigraphy consists of a coarse block layer with 50% porosity and air-filled voids which is assigned low field capacity of 1% (Table 1), i.e. the surfaces of the coarse blocks retain only little water. This idealized stratigraphy is designed to represent an active rock glacier where finer sediments resulting from weathering and erosion processes are transported towards the tongue of the rock glacier. Furthermore, Dahl (1966) observed that blockfields on slopes more often do not contain a fine sediment fraction between the blocks in northern Norway, so that the blocks only stratigraphy can also represent active blockfields. The second stratigraphy, blocks with sediment, is designed to represent blocky terrain where the voids are filled by finer sediments. This is often observed in blockfields on more flat surfaces, which are more likely to retain finer sediment within their pores (as in Isaksen et al., 2003 and Dahl 1966). We again consider coarse blocks with 50% porosity (as for the blocks only stratigraphy), but as the voids are filled with fine sediments (which again are assumed to have 50% porosity), the overall porosity is only 25%. Furthermore, a significantly higher field capacity than for the blocks only stratigraphy is assigned as more water can be held in the finer pores of the sediment fraction. Finally, the sediment only stratigraphy serves as a control scenario for a soil without blocks. It contains sediment with 50% porosity and a high field capacity due to the water holding capacity of the fine-grained sediment material. For all stratigraphies, bedrock (3% porosity and saturated conditions, e.g. Hipp et al. 2012; Fabrot et al. 2011) is assumed below 5 m depth, which is in line with observations from Isaksen et al. (2003) at Juvvasshøe. Finally, none of the stratigraphies contain soil organic matter. We emphasize that the stratigraphies are in qualitative agreement with field observations of air and sediment-filled block layers in Norway, but the assumed porosities of 50% for both the block layer and the sediments represent idealized scenarios.*

(3) We explain the reason why the snowfall factor was varied in the different model scenarios:

*Line 285-287: this site is extremely exposed to wind and most snow is blown away (Isaksen et al. 2003; Westermann et al., 2013), the snowfall factor is stepwise decreased to values below one to inprove the model performance.*

*Line 293-301: As the heavily wind-affected snow cover is a key source of spatial variability in ground temperatures in the Norwegian mountains (Gisnås et al., 2014; Gisnås et al., 2016), the model is run for a range of snowfall factors between 0.0 and 1.5 (Table 2) for each scenario. This analysis allows us to identify the magnitude of the thermal anomaly that the subsurface drainage induces at various amounts of snow, as well as estimate the threshold snow amount for permafrost existence in the six scenarios.*

**2 The effects of drainage, snow, soil moisture, etc. have been studied for other permafrost sites, but may not be particularly for mountainous regions, the physics does not change from low- and moderate-relief regions to mountainous regions. How does this study connect with existing**

**literature that studied the effect of snow, soil moisture, etc. on permafrost thermal regime? A better referencing is needed**

We restructured the introduction (see major comment 3) and added more references to improve the connection of this study to previous literature. An example with extended references is given below.

*Line 40-45: Snow is an important factor in governing ground temperatures and permafrost distribution within an area (e.g. Zhang et al., 2001; Zhang 2005; Goodrich, 1982), especially in mountain areas where permafrost is often associated with a shallow snow cover (e.g. Gisnås et al., 2014; Luetschg et al., 2004). The influence of soil moisture is complicated as it has an impact on the surface energy balance (e.g Liljedahl et al., 2011), the thermal characteristics of the soil (e.g. Göckede et al., 2017), and freezing/thawing dynamics (e.g. Hinkel et al., 2001; Hinkel and Outcalt, 1994), which can lead to both lower and higher ground temperatures.*

Furthermore, we included additional references in the discussion:

*Line 515-520: This timing of the ground ice formation is strongly different from all other scenarios, for which ground ice mostly forms in fall/early winter due to refreezing of the water contained in the active layer ( e.g. Hinkel et al., 2001). This refreezing of the active layer can take several months and is further delayed if a significant snow cover forms during this period, which leads to overall higher winter temperatures in the permafrost due to the insulation (Zhang, 2005).*

**3 I would strongly suggest rewriting/reorganizing the Introduction (also Methods) section. The authors have done a good job in providing detailed background; however, it needs to be organized so the reader can follow it. Especially, I found a disconnect between the driving mechanisms and how this work is going to address those. The last two paragraphs in the Introduction section provide a slight background but that needs to be expanded.**

We followed the suggestion of the reviewer and restructured the introduction. We start with a general section on permafrost and its controlling factors, including the subsurface stratigraphy. Second, we introduce the rock glacier landform and blocky terrain, followed by a presentation of the known controlling mechanisms for the negative thermal anomaly. This is followed by a paragraph on the representation of mountain permafrost in process models and mapping approaches, which typically lack a representation of key processes in blocky terrain and thus do not reproduce the thermal anomaly. We finish with explaining the goal of the study:

*Line 105-108: The goal of the study is to evaluate to what extent the thermal anomaly in blocky terrain can be simulated by such a comparatively simple scheme which could in principle be integrated in larger-scale permafrost modelling and mapping efforts. In particular, we investigate the interplay with the seasonal snow cover and discuss the impact on the permafrost distribution in mountain environments.*

The methods section has been completely restructured, resulting in new sub-sections to increase clarity. Section "3.1 The CryoGrid community model" contains a general description of the used model, not covering any site specifics or parameter values. We removed extensive descriptions of CryoGrid capabilities which are not used in our model setup. This section is followed by "3.2 Downscaling of model forcing", a description of the model forcing data, which gives the reader information about the time period and time resolution of the model forcing. Section "3.3 Model setup" now covers the model setup and parameters, which were previously scattered throughout the chapter. As suggested by the reviewer, we created separate subchapters for the Validation runs (3.3.1), Equilibrium runs (3.3.2) and Transient runs (3.3.3).

**4 There are lots of short (4-5 lines) paragraphs throughout the manuscript, and probably not needed and can easily be merged.**

We agree and have merged a number of short paragraphs.

**5 The example of CryoGrid processes provided (lines 134-135) is highly abstract. I am not expecting to provide all the details, but at least some details for a quick reference**

Due to the new structure of the methods, we removed the rather abstract examples in the old manuscript, which are not relevant for the presented study. This avoids confusion for readers, who are not familiar with CryoGrid.

*Line 152-154: CryoGrid is a simulation toolbox for ground thermal simulations that can be applied to a wide range of modelling tasks in the terrestrial cryosphere thanks to its modular structure (see Westermann et al., 2022 for details). It is mainly applied in permafrost environments, using the finite difference method to transiently simulate ground temperatures.*

Instead, we fully concentrate on the model configuration and setup used in this study throughout the entire Methods section.

**6 Paragraphs in the abstract? does the journal allow it, usually not seen/recommended?**

We have removed paragraphs in the abstract.

**7 There are many places where authors need to be specific. for example, L135: "Likewise, different process representations for the seasonal snow cover can be chosen." this needs to be expanded to mention specific snow processes rather than "different processes"**

As mentioned in the major comment 3 and 5, we restructured the methods and carefully revised the explanations, so that unspecific wording is avoided. Furthermore, we removed irrelevant examples (see answer to major comment 5). In contrast, we are more specific in the descriptions of the model setup applied in the presented study (see section "3.3 Model setup").

**8 Sensitivity to computational domain depth and bottom boundary condition is needed. Provide details that why the domain depth of 5 m and the prescribed geothermal flux were chosen. Describe if the results are sensitive, they will be, to the domain depth and bottom boundary conditions. I understand this can get complicated but at least mention it in the text.**

At 0-5 m depth, sediment with different characteristics (see model scenarios) are applied in the model. Below 5 m depth, bedrock is assumed. The model domain depth is at 100 m depth where a heat flux boundary condition (geothermal heat flux of 0.05 $Wm^{-2}$) is applied. A model domain depth of 100 m is a typical value in permafrost simulations focussing on decadal to centennial timescale which has already been used in previous studies in Norway (Westermann et al., 2016; Westermann et al., 2022; Schmidt et al., 2021). We clarified the differences between depth of the sediment and model domain depth in the manuscript.

*Line 154-156: We use a one-dimensional model column with a domain depth of 100 m (as in e.g. Westermann et al., 2016; Schmidt et al., 2021) and grid cell sizes increasing with depth (Fig. 2). The lower boundary condition is provided by a constant geothermal heat flux.*

*Line 232-234: For all stratigraphies, bedrock (3% porosity and saturated conditions, e.g. Hipp et al. 2012; Fabrot et al. 2011) is assumed below 5 m depth, which is in line with observations from Isaksen et al. (2003) at Juvvasshøe.*

Furthermore, we included a new figure to illustrate the model column:

[Figure]

*Line 165-168: Figure 2: Schematic of the model grid, indicating cell sizes at different depths and upper and lower boundary conditions. As upper boundary condition, the surface energy balance (SEB) forced by near-surface meteorological data is used. The lower boundary condition is provided by a constant geothermal heat flux at 100 m depth.*

**9 A schematic of the model domain with boundary conditions, soil discretization, soil layers, etc. can help better follow the results.**

We followed this suggestion and included a figure illustrating the model domain (see response to major comment 8). It gives an overview of grid cell sizes, domain depth and upper and lower boundary conditions.

**10 Figure 7 shows results for transient runs (1951-2019). What is the air temperature gradient (or increase in the mean annual air temperature) over this period? and did the authors try to run detrended data to isolate the effect of temperature increase? Otherwise, this effect is not due to soil stratigraphy and drainage only. And since the porosity in the "Blocks with sediment" case is 25% (half of the two other cases), more degradation is not unexpected. Also, what caused the patchy low ice content in some of the subplots in Figure 7 (for example, top/bottom right)?**

The air temperature change between the 1951-1960 and 2010-2019 means were 0.7 °C and 1.2 °C at Juvvasshøe and Ivarsfjorden, respectively. We included this information in the revised manuscript.

*Line 402-404: The ERA5 reanalysis dataset allows us to simulate the evolution of the ground thermal regime and ground ice content from 1951 to 2019, during which mean air temperatures increased from -4.5 °C (1951-1960) to -3.8 °C (2010-2019) for Juvvasshøe and from 0.5 °C (1951-1960) to 1.2 °C (2010-2019) at Ivarsfjorden.*

We did not run the model with a detrended dataset, as the goal is to investigate the effect of climate change (and thus air temperature increase) with the transient runs. The effect of soil stratigraphy and drainage is analyzed with the equilibrium runs.

We agree with the reviewer that more degradation is expected in the *blocks with sediment* scenarios as this stratigraphy features half of the porosity (and thus ice content) of the other two stratigraphies. We have now included this point in the text.

*Line 413-415: The complete degradation in the blocks with sediment runs compared to partial degradation in all other scenarios (except blocks only, drained) is not unexpected since this stratigraphy has a 25% porosity (and thus ice content), compared to 50% in the others.*

We thank the reviewer for pointing out the patchy lower ice content. We could explain this effect with an error in the initialization. We corrected the settings and performed the model runs again. The qualitative behavior the simulation results is the same as before, so this correction does not change any of the conclusions of the study.

[Figure]

Line 420-423: Figure 8: Modelled volumetric ground ice content at Ivarsfjorden between 1951 and 2019 for the idealized stratigraphies in undrained and drained conditions and sf = 1. The ground surface elevation is at 106 m.a.s.l. in the active layer, ice contents increase and decrease annually, corresponding to the active layer refreezing and thawing.

**11 The authors kept referring to "at depth 5 m". Please draw/highlight surface elevation (datum) in the figures. For instance, what elevation would be "5 m depth" in Figure 7? It is not clear.**

We follow the suggestion of the reviewer and highlighted the surface elevation in figure 7 and 8 (figure 6 and 7 in the old manuscript). Furthermore, we clarified the figure captions. For changes in figure 7, see response to comment 10. Changes in figure 8 are given below.

[Figure]

*Line 397-400: Figure 7: Modelled volumetric ground ice content in the upper 1 m of the ground (below 1894 m) and the snow cover (above 1894 m) for the blocks only, drained scenario, during one year of an equilibrium run at Juvvasshøe for sf = 0.75. Note the rise of the ground ice table in June after infiltrated snow melt water refreezes.*

**12 The study is performed on an idealized column domain with a fixed surface datum. So, it would be easy for the reader to have the vertical scale in "depth" [0,5], instead of elevation, which I don't think is needed unless I am missing something.**

The surface datum is indeed fixed in the model setup of this study. The forcing data is however downscaled for the exact surface elevations of the two sites. Since figure 7 shows both the snow cover above the surface and ground ice below the surface, we remain consistent and have elevation on the vertical scale. For clarity, we highlighted the surface elevation in both figure 7 and 8 (see major comment 10 and 11).

**13 The focus of this work is to study the effect of soil stratigraphy (with drained/undrained conditions) on the soil thermal regime; however, no sensitivity study is performed on "soil stratigraphy". For instance, how the blocks only scenario with porosities of 0.6 and 0.4 will affect permafrost conditions?**

We performed a sensitivity study on different soil stratigraphies. The Supplement now contains results from additional simulations with a porosity of 0.4 and 0.6. For *blocks with sediment*, both the porosity of the blocks and of the sediment have been adjusted accordingly. The results show that ground temperatures at 2 m depth are within 0.4 °C between porosity 0.6 and 0.4. This means that the general results and conclusions are not influenced by small porosity changes.

*Line 11-18 (in Supplement): Here, we provide the sensitivity of mean ground temperatures at 2 m depth for differences in porosity in the blocky layer (upper 5 m of the ground). Simulations are setup as in section 3.3.2 (equilibrium runs) but for three different porosities at a single snowfall factor. For blocks with sediment stratigraphy, we assume the porosity value for each the blocks and the sediment to be the same. For example, with porosity 0.4, blocks with 40% porosity, which are filled with sand which also has 40% porosity, resulting in a final porosity of 0.16 (0.32 for porosity 0.6).*

**Suppl. 2, Table 1: Equilibrium ground temperature (°C) at 2 m depth for the three idealized stratigraphies at three different porosities. The snowfall factors are the same as resulted from the model validation.**

| Site | Stratigraphy | Porosity 0.4 | Porosity 0.5 | Porosity 0.6 |
|---|---|---|---|---|
| **Juvvasshøe** | *Blocks only* | -3.2 | -3.2 | -3.1 |
| | *Blocks with sediment* | -3.0 | -3.2 | -3.4 |
| | *Sediment only* | -3.7 | -3.6 | -3.4 |

| | | | | |
|---|---|---|---|---|
| **(sf = 0.25)** | | | | |
| **Ivarsfjorden** | *Blocks only* | -0.3 | -0.4 | -0.2 |
| | *Blocks with sediment* | 2.0 | 1.8 | 1.6 |
| | | | | |
| **(sf = 1)** | *Sediment only* | 1.8 | 1.6 | 1.5 |

Furthermore, the sensitivity towards the parameter d$^{lat}$ has been tested and is presented in the supplement.

**14 I would also suggest some of the sensitivity-related results, for example, section 4.2, to be moved to a supplemental document, and focus more on what is new here.**

The insulating effect of the snow cover on the thermal regime has been studied before and is not a new finding in this study. However, the sensitivity towards the snowfall factor is an important finding in this study which we would like to keep in the main manuscript. Spatial variability of the snow cover is a typical phenomenon in the Norwegian mountains, and it is a key control for the existence of permafrost. In this study, we show that the snow threshold for permafrost existence strongly depends on the subsurface stratigraphy. Therefore, we analyze the magnitude of the thermal anomaly that the subsurface drainage induces at various amounts of snow and estimate a snow threshold for permafrost existence in the different model scenarios. To increase clarity, we made the following changes in the methods:

*Line 297-301: As the heavily wind-affected snow cover is a key source of spatial variability in ground temperatures in the Norwegian mountains (Gisnås et al., 2014; Gisnås et al., 2016), the model is run for a range of snowfall factors between 0.0 and 1.5 (Table 2) for each scenario. This analysis allows us to identify the magnitude of the thermal anomaly that the subsurface drainage induces at various amounts of snow, as well as estimate the threshold snow amount for permafrost existence in the six scenarios.*

and in the results:

*Line 356-359: For snowfall factors of 0.75 and larger, the difference in ground temperature between blocks only, drained and the other scenarios is in the range of 1.1 °C and 1.8 °C at Juvvasshøe and in the range of 1.1 °C and 2.2 °C at Ivarsfjorden. This shows that the magnitude of the negative thermal anomaly increases with a larger amount of snowfall.*

We also emphasized the importance of the sensitivity tests to snowfall factors in the abstract and conclusions:

*Line 24-25: The thermal anomaly increases with larger amounts of snowfall, showing that well drained blocky deposits are less sensitive to snow insulation than other soils.*

*Line 589-591: The largest anomalies occur in simulations with a thick winter snow cover as ground temperatures in well drained blocky deposits are less sensitive to insulation by snow than other soils.*

Finally, we removed the results of the sensitivity warming rates at different snowfall factors as this does not add enough to the objective of the study.

**15 Also, in section 4.1 (line 307) authors mentioned all simulations used a single snowfall factor, however, later in section 4.3 they use different snowfall factors. This needs more explanation. Providing a table (which may not be in the main manuscript) listing all scenarios (other than the three listed in Table 1) will help the reader better understand it otherwise it is hard to untangle.**

We agree that it was previously unclear which snowfall factors were applied in the different model scenarios. The revised manuscript includes a table, which gives the reader an overview of the simulations and applied settings, including snowfall factors.

*Line 265-268: Table 2: Overview of basic model settings for the different simulation types. A spin-up of subsurface temperatures is achieved by repeated simulations for the spin-up period (until a stable temperature profile is reached), before the actual model run for the simulation period is conducted. "Idealized" stratigraphy and drainage refers to three subsurface stratigraphies (Table 1) combined with two types of drainage conditions. See Sect. 3.3.1 to Sect. 3.3.3 for details.*

| Run type | Site | Spin up | Simulation | Stratigraphies | Snowfall factors |
|---|---|---|---|---|---|
| *Validation* | Juvvasshøe | 1951-2010 | 2010-2019 | Best-fit | 0.25 |
| | Ivarsfjorden | 1951-2016 | 2016-2019 | Best-fit | 1 |
| *Equilibrium* | Juvvasshøe | 2000-2010 | 2000-2010 | Idealized | 0.0, 0.25, 0.5, 0.75, 1.0, 1.5 |
| | Ivarsfjorden | 1962-1971 | 1962-1971 | Idealized | 0.0, 0.25, 0.5, 0.75, 1.0, 1.5 |
| *Transient* | Juvvasshøe | 1962-1971 | 1951-2019 | Idealized | 0.25 |
| | Ivarsfjorden | 1962-1971 | 1951-2019 | Idealized | 1 |

**16 What type of soil retention curve is used in the study? Moreover, how sensitive are the results to hydraulic conductivity? I didn't find any mention of the role of spatial gradient (steepness). All these factors significantly impact drainage. Discuss.**

We use a gravity driven bucket scheme, where water in excess of the field capacity infiltrates downwards. We clarified this in the revised manuscript:

*Line 169-173: For soil hydrology, a gravity driven bucket scheme is used (Westermann et al 2022). Rainfall provided by the model forcing is added to the uppermost grid cell, while evaporation is determined by the surface energy balance calculations (note that we consider unvegetated surfaces and thus do not account for transpiration). Water that is in excess of the field capacity infiltrates downwards until either the water table or a non-permeable layer, such as a frozen grid cell is reached. If all grid cells are saturated, excess water is removed as surface runoff.*

The description of the seepage face and the corresponding drainage has been revised and provides now a better overview of the influencing factors:

*Line 173-185: We use a one-dimensional model setup, but simulate lateral drainage of water by introducing a seepage face, i.e. a lateral boundary conditions for water fluxes representing flow between saturated grid cells of the model domain and a stream channel (or the atmosphere) to which the water can freely flow out from the subsurface (e.g. Scudeler et al., 2017). Using the elevation of the water table, $z_{wt}$ (computed as the elevation of the uppermost saturated grid cell), lateral water fluxes $F_i^{lat}$ are derived for all saturated unfrozen grid cells i below the water table (i.e. at elevations $z_i < z_{wt}$ ) as*

$$F_i^{lat} = -K_H \frac{z_{wt} - z_i}{d^{lat}} \, ,$$

where $K_H$ is the *saturated* hydraulic conductivity, $d^{lat}$ *is the lateral distance to the seepage face and the flux is determined by the difference between the hydrostatic potential (proportional to $z_{wt}$) of the water column and the gravitational potential of free water at the elevation of each cell (proportional to $z_i$). Note that Eq. (1) is an approximation for small changes of the water table and small outflow fluxes for which the potential in the saturated zone can be approximated by the hydrostatic potential. The parameter $d^{lat}$ is used to control the strength of the drainage, with small distances resulting in a well-drained column, while high values lead to suppressed drainage.*

In our study, we investigate only two extreme cases of essentially no drainage and very good drainage. A better description of the setup has been included in the manuscript:

*Line 245-255:* *To investigate the effect of subsurface drainage on ground temperatures and ground ice conditions, we distinguish undrained and drained scenarios by using two different values of $d^{lat}$ (Eq. 1) for in the idealized stratigraphies. A $d^{lat}$ value of $10^4$ m is used for undrained cases, which emulates conditions at a flat surface, resulting in a to a good approximation one-dimensional water balance, where only surface water is removed. For the drained cases, a $d^{lat}$ value of 1 m is used, which results in well-drained conditions which are typical in sloping terrain. For the saturated hydraulic conductivity $K_H$, a fixed value of $10^{-5}$ m s$^{-1}$ is used for all stratigraphies, although the true hydraulic conductivities almost certainly differ between stratigraphies. However, the key parameter controlling lateral water fluxes in Eq. 1 is in reality the "drainage timescale" $K_H/d^{lat}$ [s$^{-1}$], which is varied by four orders of magnitude between $K_H/d^{lat}$ = $10^{-5}$ s$^{-1}$ ($d^{lat}$ = 1 m, well-drained conditions) and $K_H/d^{lat}$ = $10^{-9}$ s$^{-1}$ ($d^{lat}$ = $10^4$ m undrained conditions). As the study setup is designed to analyze these two "confining cases", it is sufficient to only vary $d^{lat}$ and leave $K_H$ constant for simplicity. Further sensitivity tests for $d^{lat}$ and $K_H$ are provided in the Supplement.*

In addition, we performed a sensitivity analysis to $d^{lat}$ in the Supplement.

*Line 20-27 (in Supplement):* *Here, we provide the sensitivity of mean ground temperatures at 2 m depth for differences in d$^{lat}$, which is the parameter used to control the drainage rate. Simulations are setup as in section 3.3.2 (equilibrium runs) but with five different values for d$^{lat}$ and one snowfall factor. The increase of d$^{lat}$ by one order of magnitude results in the same drainage rate as decreasing the K$_H$ (saturated hydraulic conductivity) by one order of magnitude (see Eq. 1). sf = 1 is used, as differences between drainage rates are minimal for sf = 0.25 (Fig. 4).*

**Suppl. 3, Table 1: Equilibrium ground temperature (°C) at 2 m depth for the three idealized stratigraphies at five values of $d^{lat}$.**

| Site | Stratigraphy | $d^{lat}$ $10^4$ m | $d^{lat}$ $10^3$ m | $d^{lat}$ $10^2$ m | $d^{lat}$ $10^1$ m | $d^{lat}$ $10^0$ m |
|---|---|---|---|---|---|---|
| **Juvvasshøe** | *Blocks only* | 0.3 | 0.0 | 0.0 | -0.7 | -0.9 |
|  | *Blocks with sediment* | 0.3 | 0.3 | 0.3 | 0.2 | 0.2 |
| **(sf = 1)** | *Sediment only* | 0.3 | 0.3 | 0.3 | 0.2 | 0.0 |
| **Ivarsfjorden** | *Blocks only* | 1.3 | 0.3 | 0.1 | -0.1 | -0.4 |
|  | *Blocks with sediment* | 1.8 | 1.8 | 1.8 | 1.8 | 1.8 |
| **(sf = 1)** | *Sediment only* | 1.7 | 1.7 | 1.7 | 1.6 | 1.6 |

**Minor comments:**

**1. Line 13: effect of drainage on what? soil thermal regime? please explain**

We added the explanation.

*Line 13-15: Here we used the CryoGrid community model to simulate the effect of drainage on the ground thermal regime and ground ice in blocky terrain permafrost at two sites in Norway*

**2: Line 15: please explain here what type of model domain was considered? Is it a 2D generic hillslope or observed 2D/3D topography?**

We added that model domain is one-dimensional.

*Line 15-16: The model setup is based on a one-dimensional model domain and features a surface energy balance, heat conduction and advection, as well as a bucket water scheme with adjustable lateral drainage.*

**3. please clarify what does drained/undrained mean here. saturated or unsaturated conditions?**

We shortly explained what differentiates the drained and undrained scenarios.

*Line 16-18: We used three idealized subsurface stratigraphies, blocks only, blocks with sediment and sediment only, which can be either drained (i.e. with strong lateral subsurface drainage) or undrained (i.e. without drainage), resulting in six scenarios.*

**4. remove quotes**

Agreed and the quotes have been removed.

**5. I assume it means, there is more permafrost in the blocks only, drain case. How much in terms of percentage as compared to other two cases?**

We rephrased this sentence in order to include a quantitative statement about ground ice loss. These percental decreases in ground ice content reflect the volumetric loss of ground ice between the active layer and the top of the bedrock. These numbers are also included in the results chapter.

*Line 27-30: Finally, transient simulations since 1951 at the rock glacier site (starting with permafrost conditions for all stratigraphies) showed a 100% lowering of the ground ice table in the blocks with sediment, drained run, 37% lowering in the sediment only run and only 2% lowering in the blocks only, drained run.*

and

*Line 408-413: In the blocks only, drained scenario, the perennial ice table in the upper 5 m (so between the active layer and the bedrock) does not lower by a significant amount (2 % lowering), while the ice table lowers by 33 % in the blocks only, undrained scenario. The ice table in the blocks with sediment stratigraphy disappeared by 1985 and 1975 in the undrained and drained scenarios respectively. Finally, the sediment only simulations show an intermediate effect where the ice table has lowered by 41 % and 39 % for undrained and drained conditions respectively by 2019.*

**6. would be nice to provide a rough number here, like what is that limit you are *referring* to?**

We included estimations of elevational permafrost limits from (Etzelmüller et al., 2003; Gisnås et al., 2017) for the region of the two study sites in the Introduction.

*Line 50-52: In Southern Norway, the lower limit of mountain permafrost is estimated between 1600 m a.s.l. in the west to 1000 m a.s.l. in the east (Etzelmüller et al., 2003), while a similar west-east decrease from 800–1000 m a.s.l. to ca. 300 m a.s.l. in the east is observed in Northern Norway (Gisnås et al., 2017).*

**7. this effect? this is unclear. does it refer to the role of heat conduction, which is 'mostly' included in permafrost simulation, or does it say 'consider simulating permafrost below that assumed elevational limit?**

We clarified that that it regards the subsurface water/ice balance.

*Line 32-33: It is thus important to consider the subsurface water/ice balance in blocky terrain in future efforts on permafrost distribution mapping in mountainous areas*

**8. be specific here. not all mountain environments have permafrost**

We include a clarification regarding the high latitude and/or high altitude of mountain permafrost environments

*Line 36-38: Permafrost is defined as ground that remains at or below 0 °C for two or more consecutive years (Van Everdingen, 1998) and is a common feature at high elevations and/or mid-latitude mountain environments, where permafrost occurs even in mid- and low-latitudes (Gorbunov, 1978)*

**9.for a smooth flow, at least mention different types of permafrost here, such as continuous, discontinuous, sporadic.**

We included an extra sentence that explains the different classifications of permafrost regarding aerial extent.

*Line 38-40: Different permafrost zones are classified based on the aerial extent of permafrost presence. These zones are: continuous, discontinuous, sporadic and isolated, where the surface in underlain by permafrost in more than 90%, 50-90%, 10-50% and less than 10% of the land area respectively (Smith and Riseborough 2002).*

**10. again, what is that assumed elevation limit**

We included estimations of elevational permafrost limits from (Etzelmüller et al., 2003; Gisnås et al., 2017) for the region of the two study sites in the Introduction.

*Line 50-52: In Southern Norway, the lower limit of mountain permafrost is estimated between 1600 m a.s.l. in the west to 1000 m a.s.l. in the east (Etzelmüller et al., 2003), while a similar west-east decrease from 800–1000 m a.s.l. to ca. 300 m a.s.l. in the east is observed in Northern Norway (Gisnås et al., 2017).*

**11. not clear if this is authors' analysis of Juliussen and Humlum work or they are just refering to their work.**

The Introduction has been restructured for clarity, it is now clear that we refer to the work of Juliussen and Humlum.

**12. please provide some specific examples of applications**

A selection of studies that were performed with CryoGrid are listed in order to provide the reader examples of previous work with the model.

*Line 95-99: It largely builds on the well-established CryoGrid 3 model (Westermann et al., 2016), which has been used in e.g. peat plateaus and palsas (Martin et al. 2021), ice-wedge polygons (Nitzbon et al. 2019)*

*and boreal forests (Stuenzi et al. 2021) and has a broad range of applications, including the representation of lateral drainage regimes (Martin et al., 2019), representation of steep rock walls (Schmidt et al., 2021) and massive ice bodies.*

**13. rephrase**

The entire paragraph was rephrased.

**14. mean annual ground?**

This value of -2.5 °C covers the entire period of 2000 and 2004. We use the term mean ground temperature in the same way as the source of this value (Isaksen et al., 2007).

**15. mean annual? I see the unit here is mm/yr (rate), but general practice is to provide "mean" in mm (amount)**

Agreed and adjusted.

*Line 118-119: The mean annual precipitation was estimated to be between 800 and 1000 mm.*

**16. I guess, Mean annual ground surface temperature? this is never defined before. please define it before using it.**

We included the definition

*Line 121-123: Isaksen et al. (2007) measured the difference between the mean annual ground surface temperature (MAGST) and mean annual air temperature (MAAT), which is the surface offset at exposed and less exposed sites in this area.*

**17. and how deep is the permafrost table? i.e., depth of permafrost from the surface at the exposed sites and/or sites with significant snow cover. If you have this information available it would be good to add.**

We included information about the active layer thickness (and thus depth to the permafrost table).

*Line 125-127: The thickness of the active layer increased from 215 cm in 1999 (Isaksen et al., 2001) to ca. 250 cm in 2019 (Etzelmüller et al., 2020).*

**18. it needs to be defined, if it has not been defined already**

See comment 15, it has now been defined before.

**19. how is this depth picked for constant water/ice content layer? and how is this depth related to the damping depth of the surface thermal signal?**

This paragraph was completely rephrased. The comment referred to an example in the old manuscript, which was not relevant for the presented study. It is deleted in the new manuscript (see as well response to major comment 5).

**20. this also needs some explanation. Like what type of snow processes? aging, compaction, density variations etc.? It would be hard for someone not familiar with CryoGrid to follow it.**

This paragraph was completely rephrased. The comment referred to a rather abstract example in the old manuscript, which was not relevant for the presented study. It is deleted in the new manuscript (see as well response to major comment 5 and minor comment 20).

**21. how do you compute the water table location? Explain**

The water table is computed by taking the uppermost saturated grid cell. We clarified that in the description of the drainage scheme.

*Line 176-177:* *Using the elevation of the water table $z_{wt}$ (computed as the elevation of the uppermost saturated grid cell), lateral water fluxes…*

**22. above this water table?**

Water below the water table is removed (and thus the water table lowers). Water above the water table is held by the field capacity or infiltrates downwards and is thus not removed with the drainage component. We clarified this in the manuscript:

*Line 173-178: We use a one-dimensional model setup, but simulate lateral drainage of water by introducing a seepage face, i.e. a lateral boundary conditions for water fluxes representing flow between saturated grid cells of the model domain and a stream channel (or the atmosphere) to which the water can freely flow out from the subsurface (e.g. Scudeler et al., 2017). Using the elevation of the water table, $z_{wt}$ (computed as the elevation of the uppermost saturated grid cell), lateral water fluxes $F_i^{lat}$ are derived for all saturated unfrozen grid cells i below the water table (i.e. at elevations $z_i < z_{wt}$ ) as…*

**23. How is this 5 m depth picked or consistent with observations?**

At Juvvasshøe, bedrock is observed at 5 m depth. We included a clarification. We do not have observations at the rock glacier in Ivarsfjorden. An estimate of 5 m is reasonable and most importantly assures consistency between the model runs.

*Line 232-234: For all stratigraphies, bedrock (3% porosity and saturated conditions, e.g. Hipp et al. 2012; Fabrot et al. 2011) is assumed below 5 m depth, which is in line with observations from Isaksen et al. (2003) at Juvvasshøe.*

**24. if it is zero for all cases, it can be removed from the Table and just mention it in the main text.**

We agree and have removed it from the table and included it in the main text

*Line 234-235: Additionally, none of the stratigraphies contain soil organic matter.*

**25. have a separate column for the "three cases" instead of putting it under the "Depth" column**

We agree, the table is adjusted.

**26. Seems to me that Figure 2 does not support this statement, as at depth 4 m, the model fails to follow the observed trend particularly in the spring shoulder season for all years. Is it due to bottom boundary condition or snow model or thermal conductivities? How the authors ensured that the simulations won't be affected by the bottom boundary condition if a 5 m domain is used?**

We assume the referee refers to a depth of 2 m. To address this and other comments, we have included a statistical analysis of model fit. At both of the sites, the bias and RMSE are calculated with daily values.

*Line 291-293: At both sites, the root-mean-square-error (RMSE) and bias are calculated in order to provide an objective measure of the model fit. At Juvvasshøe this was accomplished for daily values at 0.4 m and 2 m depth, while at Ivarsfjorden the mean daily ground surface temperature of the loggers within the rock glacier outline is used.*

*Line 328-330: This configuration used drained conditions, although differences with undrained conditions are minimal for this stratigraphy. For daily temperatures at 0.4 m depth, the RMSE and bias are 2.1 °C and -0.6 °C, respectively, while they are 1.2 °C and -0.7 °C at 2 m depth.*

*Line 341-342: The best-fitting model configuration was found to be the blocks with sediment stratigraphy and a snowfall factor of 1.0, resulting in an RMSE of 1.3 °C and a bias of -0.4 °C.*

Furthermore, we added a possible explanation for the mismatch of spring temperatures:

*Line 330-335: There is a mismatch in the timing of spring temperatures at 2 m depth in several years, for which modelled temperatures increase later than measured values. This is likely a result of differences in the snow melt, as the snowpack dynamics resulting from wind redistribution is not completely captured by the snowfall scaling with a constant snowfall factor (e.g. Martin et al., 2019). Furthermore, the uppermost 1 m contain large stones and boulders, while the layer below is characterized by smaller stones and cobbles (Isaksen et al. 2003), so that a ground stratigraphy with two layers in the uppermost 5 m may further improve the performance of the simulations.*

The model domain depth is 100 m, so that the lower boundary conditions does not affect the results in this study (see response to major comment 8).

**27. how well modeled MAGST matches the observed for the other site?**

At Juvvasshøe, ground surface temperatures are not monitored, but we have near-surface measurements which are equally suited for model validation. The analysis/comparison is done at 0.4 m depth, and at 2 m depth.

**28. I found it confusing. Capturing general trends in the ground surface temperature is no guarantee that soil thermal state is represented accurately due to the complex and nonlinear nature of the subsurface.**

We agree with the reviewer and added a statistical analysis (RMSE, bias) for daily ground surface temperatures at Ivarsfjorden:

*Line 291-293: At both sites, the root-mean-square-error (RMSE) and bias are calculated in order to provide an objective measure of the model fit. At Juvvasshøe this was accomplished for daily values at 0.4 m and 2 m depth, while at Ivarsfjorden the mean daily ground surface temperature of the loggers within the rock glacier outline is used.*

Furthermore, we present a new figure with daily resolution for Ivarsfjorden:

[Figure]

*Line 349-352: Figure 4: Daily modelled and measured ground surface temperatures in Ivarsfjorden from July 2016 to July 2019. The shaded area indicates the minimum to maximum range of measured daily values from 11 loggers (based on Lilleøren et al., 2022), while the black line represents the mean value of all loggers.*

Unfortunately, no boreholes with ground temperatures measurements exist in rock glaciers of similar interest in Norway. We therefore use 11 temperature loggers, which provide a good overview of the ground surface temperatures of Ivarsfjorden.

**29. highlight the summer reason in Fig. 5**

We highlighted the snow-free summer period in the figure.

[Figure]

*Line 385-388: Figure 6: Modelled ground temperature at 0.05 m (top) and 2 m (bottom) depth for the blocks only, drained and blocks with sediment, drained scenarios during a year of an equilibrium run at Juvvasshøe for sf = 0.75. The snow-free summer season is highlighted. Note that the upper plot is truncated at 17 °C, maximum summer temperatures are 26 °C in both scenarios.*

**30. The upper plot seems truncated vertically. Either adjust it or mention it in the caption**

We mentioned it in the caption.

*Line 387-388: Note that the upper plot is truncated at 17 °C, maximum summer temperatures are 26 °C in both scenarios.*

**31. This needs more details, dry soils have low thermal conductivity, so how can it enable rapid refreezing? There is always a competition between the soil thermal conductivity and heat capacity.**

We added a clarification on that in the revised manuscript:

*Line 511-513: Dry soils have a lower thermal conductivity compared to wet soils, but the lack of latent heat release allows for rapid refreezing during fall which enables fast cooling of the deeper soil layers and thus leads to overall lower winter temperatures.*

**32. 1. What is the surface elevation? highlight it on the plots. 2. I wonder how was this model initialized that the authors got low water content around 104 m and above/below the water content is very different and high? 3. Same for bottom right plot. what is causing the patchy low water content areas?**

1. The surface elevation is 106 m, which is the top of the plot. In order to clarify that this figure shows only the subsurface, we added to the figure caption and highlighted the surface in the plot. We did the same for other plots of ground ice and snow cover (see major comment 11).

2.& 3. The model is initialized with saturated conditions and an error in the simulation setup was corrected, removing the patchy ice contents (major comment 10).

**33. Provide more details in the caption and letter the subplots for easy referencing.**

We lettered the subplots for easier referencing. Furthermore, we provided more details in the caption:

*Line 421-423: Figure 8: Modelled volumetric ground ice content at Ivarsfjorden between 1951 and 2019 for the idealized stratigraphies in undrained and drained conditions and sf = 1. The ground surface elevation is at 106 m a.s.l. in the active layer, ice contents increase and decrease annually, corresponding to the active layer refreezing and thawing.*

**34. This needs to be explored, that how much of subsurface warming is due to stratigraphy, soil moisture, and how much the change in the air temperature caused over the period 1951-2019, while keeping the effects of snow isolated as well.**

It is not really possible to separate these effects in a highly coupled model like CryoGrid. Changes in air temperature, for example, also impact the snow cover through winter melt episodes which again change the insulating effect of the snow cover. As this study clearly shows, the effect on ground temperatures is then once again modulated by the subsurface ice dynamics, creating a system where all variables are coupled at least to a certain degree. While it would in principle be possible to perform experiments where the snowfall forcing of e.g. the first decade is looped (while all other parameters evolve as delivered by the ERA-5 reanalysis), we do not think that this adds anything to the analysis – also in the climate system (and the ERA-5 data which is a best-guess representation of it), there are correlations between snowfall and air temperature, so we would in this case investigate a highly artificial system which is difficult to relate to reality.

**35. how this connects with the abstract and results? It tells that the main focus of this study was to investigate the effect of snowfall on the soil thermal regime, which I don't think is the case.**

We agree with the reviewer and rephrased the sentence.

*Line 581-584: In this study, we used the CryoGrid permafrost model to simulate the effect of blocky terrain on the ground thermal regime and ground ice dynamics at two Norwegian mountain permafrost sites (Juvvasshøe and Ivarsfjorden rock glacier). In particular, we investigated the effect of subsurface drainage, as typical on slopes, for three idealized stratigraphies, named blocks only, blocks with sediment and sediment only.*